# Spatial coordination in a mutually beneficial bacterial community enhances its antibiotic resistance

Lingjun Li [1], Tian Wu[2], Ying Wang[2], Min Ran[2], Yu Kang [3], Qi Ouyang[1,2,4] & Chunxiong Luo [1,2]

Microbial communities can survive in complex and variable environments by using different cooperative strategies. However, the behaviors of these mutuality formed communities remain poorly understood, particularly with regard to the characteristics of spatial cooperation. Here, we selected two *Escherichia coli* strains, designated as the nutrition provider and the antibiotic protector, respectively, for construction of a mutually beneficial bacterial community that could be used to study these behaviors. We found that in addition to the functional mutualism, the two strains also cooperated through their spatial distribution. Under antibiotic pressure, the bacterial distribution changed to yield different spatial distributions, which resulted in community growth advantages beyond functional cooperation. The mutualistic behavior of these two strains suggested that similar communities could also use variations in spatial distribution to improve their survival rates in a natural environment or under the action of antibiotics.

[1] Center for Quantitative Biology, Academy for Advanced Interdisciplinary Studies, Peking University, Beijing 100871, China. [2] The State Key Laboratory for Artificial Microstructures and Mesoscopic Physics, School of Physics, Peking University, Beijing 100871, China. [3] CAS Key Laboratory of Genome Sciences and Information, Beijing Institute of Genomics, Chinese Academy of Sciences, Beijing 100029, China. [4] Peking-Tsinghua Center for Life Sciences, Peking University, Beijing 100871, China. Correspondence and requests for materials should be addressed to C.L. (email: pkuluocx@pku.edu.cn)

The exploration of ecosystem characteristics at the community level through studies of intraspecific and interspecific interactions is a major research direction in ecosystem biology[1–3]. Among different ecosystems, microbial systems play an important role in human health and the global nutrition cycle[3–6]. Moreover, they are sufficiently tractable for use in rapid and controlled laboratory experiments[7,8]. Therefore, microbial systems are generally selected as the research object in the study of an ecosystem[7,9,10]. Many types of interspecific cooperative relationships in microbial communities, such as cross-feeding and cross-protection, can generate mutualism, and previous studies have focused on these two types of mutualistic behavior[1,8,11,12]. A given microbial ecosystem will have different characteristics in different environments, and the interactions between populations are affected by environmental factors[12–15]. For example, bacteria that can provide nutrients and those that can degrade antibiotics may be mutualistic in a nutrient-deficient environment in the presence of antibiotics. However, when the antibiotic pressure is removed, the growth of the former bacteria does not depend on the latter, at which time there may be a parasitic relationship between the two bacteria[13]. Compared with simple cooperation, such as providing nutrition or degrading antibiotics, multitype cooperative behavior is more complex and closer to the real situation under natural conditions[16–18]. However, the available knowledge of such complex mutualism remains insufficient[7,17].

Besides functional interactions, the spatial distribution of bacteria can also affect the stability of their communities[12,19,20]. Kim et al. generated a mutualistic community of three bacterial strains, each of which was capable of either degrading antibiotics, providing a carbon source or providing a nitrogen source. A community could not develop when the three strains were uniformly mixed and cultured together, with maximal growth only achieved when they were separated by a specific distance[21]. However, due to the limitations imposed by experimental conditions, previous studies on the characteristics of the mutualism were mainly restricted to analyses of the bacterial species, quantity, and proportion in a community and did not explore the function of the spatial distribution and the dynamic changes in the spatial distribution[10,14,15,22].

Compared with a homogeneous system, microbial communities in nature, such as soil and intestinal flora, are more likely to grow in heterogeneous environments with respect to nutrients, oxygen, antibiotic pressure and other factors[19,20,23,24]. Many of these communities have limited space for growth, show poor bacterial motility within the community, often form aggregates and communicate with the outside world through limited contact surfaces[20]. The use of microfluidic chips can provide a well-defined growth environment for microbial community studies[23,25]. For example, using hydrogel in a chip can create a zero-flow environment in which substances can be exchanged[25]. These characteristics suggested that a space enclosed by gel barriers in a chip could be designed to simulate the natural environment in which bacterial communities grow. In this report, we established a microbial mutualism formed by two strains that perform the cooperative functions of feeding and protection. Using a diffusion-based microfluidic system, we studied the evolutionary trend of the spatial distribution and its function in the growth of the mutualistic population in antibiotic environments.

## Results

### Feeding-protection mutualism and diffusion-based culture system.
To establish a feeding-protection mutualism, we selected two strains of *E. coli*: one strain contains lac genes, which can utilize lactose and secrete intermediate products (such as succinic acid), and the other strain expresses a $\beta$-lactamase enzyme, which inactivates the antibiotic ceftriaxone sodium (CRO) but cannot utilize lactose[17]. The strain that can utilize lactose, MG1655, contains an additional plasmid encoding red fluorescent protein (RFP) as an indicator and is hereinafter referred to as nutritional bacteria. The other strain is a lactose-deficient strain (DH5$\alpha$) that contains a plasmid encoding green fluorescent protein (GFP) and a $\beta$-lactamase enzyme, which is hereinafter referred to as the resistant bacteria (Supplementary Table 1). Therefore, in a culture medium containing lactose as the only carbon source and the antibiotic CRO, the resistant bacteria can degrade the antibiotic in the environment to protect the nutritional bacteria, whereas the nutritional bacteria can provide nutrients to feed the resistant bacteria, allowing the two strains to establish a mutualism (Fig. 1a).

To verify the mutualistic relationship between the two strains, we conducted a control experiment in which the two strains were cultured alone or cocultured in a specific medium in 96-well plates[13]. The results showed that the cultures with only one strain did not grow, whereas the cocultures exhibited growth (Fig. 1b), indicating that these two strains form a mutualism under certain experimental conditions[16]. In addition, by designing another control experiment, we eliminated the possibility that the two strains are able to achieve mutualism by means other than the designed functions (Supplementary Tables 1–3, Supplementary Fig. 1). We speculated that the two studied strains with different functions may form a heterogeneous spatial distribution based on chemical diffusion, chemical consumption by bacteria and inhomogeneous bacterial growth within a growth space (Fig. 1a). To construct a natural environment for bacterial cultures, a diffusion-based microfluidic chip (Fig. 1c–h) was designed[26,27]. The square area in the middle of the chip (side length of 1600 μm and a height of ~18 μm) was used to culture the bacteria, and the culture medium was stored in four large reservoir wells that were connected to the culture area by fence-like channels filled with gel (height of ~4 μm) (Fig. 1c–e). Compared with the chip used in other study, our device only had source wells and the concentration gradient would change with the bacteria growth. Our system is more similar to natural conditions in that the bacterial mutualistic community aggregates and communicates with the outside world through limited contact surfaces (source) and the bacteria create a chemical gradient within the community by themselves[25]. The bacteria could be loaded from the bacterial inlet into the culture area through a pitchfork-like narrow channel (width of 20 μm and a height of 18 μm) and initially created a uniform distribution (Fig. 1g). Subsequently, the inlet and outlet for bacterial loading were blocked with gel to create a zero-flow bacterial culture area. The culture medium in the reservoir area could be replaced using a pipette to rapidly replace the surrounding media without affecting the bacterial distribution in the observation area (Fig. 1c). The culture media were renewed every 10 h to maintain constant lactose and CRO concentrations in the source wells. To observe the diffusion characteristics of the substance in the chip, in addition to the simulation calculation, we also used fluorescein for the diffusion experiment, which has a diffusion coefficient that is close to that of lactose and CRO (see the "Methods" section). The results of the fluorescein diffusion experiment and the MATLAB simulation showed that the substances added to the reservoir areas could diffuse uniformly in the culture area within ~2 h (Supplementary Fig. 2).

We verified the feasibility of establishing mutualism in the chip through a validation experiment in which the nutritional and resistant bacteria were added to the chip alone or in combination and cultured at 37 °C and the specific medium (M9 medium containing 2000 μg mL$^{-1}$ CRO and 0.8% lactose,

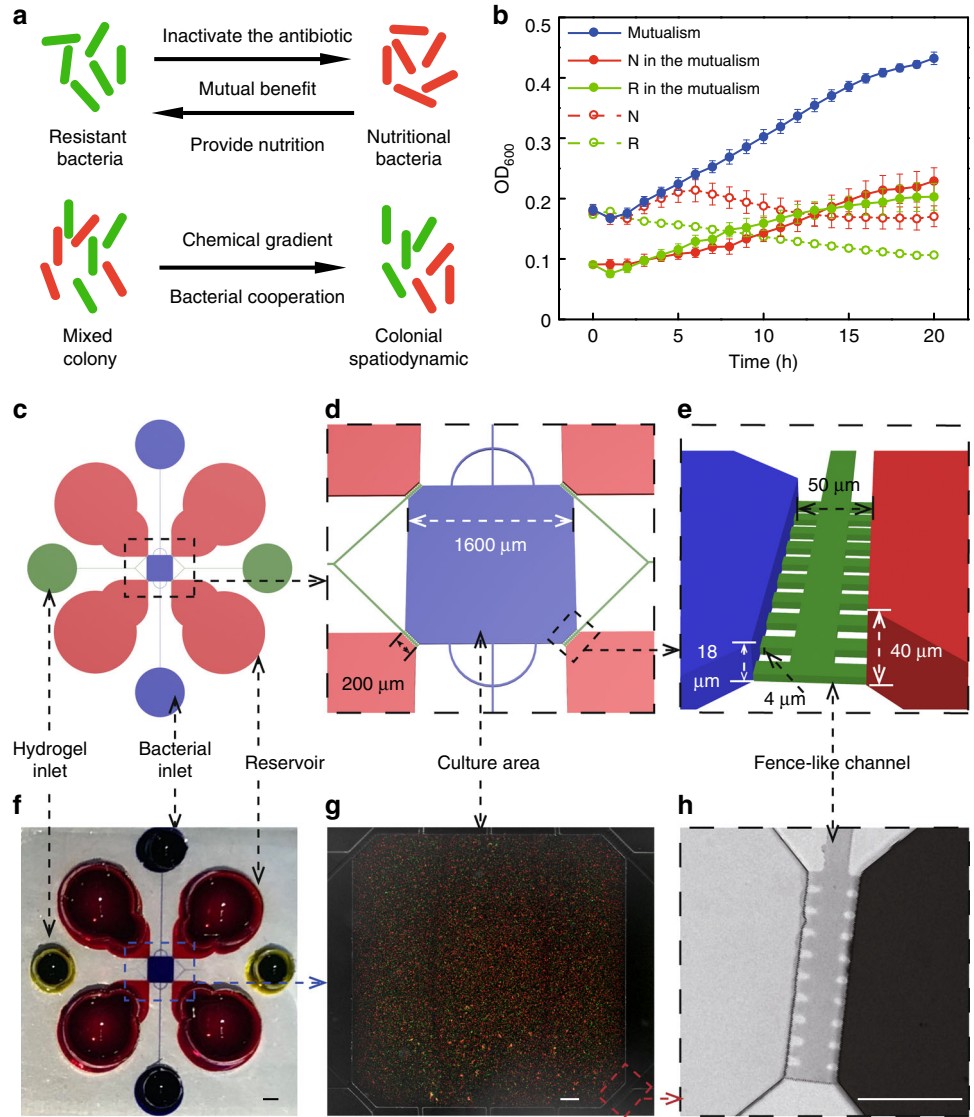

**Fig. 1** Establishment of a mutually beneficial community and design of a diffusion-based culture chip. **a** The resistant bacteria can degrade antibiotics to protect the nutritional bacteria, and the nutritional bacteria can provide nutrients for the resistant bacteria. The two bacterial strains mixed in a certain space might form a spatial distribution. **b** The microplate reader results showed that in the M9 medium containing lactose as the only carbon source and 100 μg mL$^{-1}$ CRO, only the cocultures of the two bacterial strains grew well, and no growth was observed in the single-strain cultures. The dots showed the averages of seven to eight repeated experiments and the whiskers indicated the standard deviation. The experiment was performed in 96-well plates. **c–e** Schematic images of the microdevice. The views of the device are sequentially enlarged from left to right. **f** A Photograph of the microdevice. Scale bar: 1000 μm. **g** A photomicrograph that was taken after the bacteria were added but before the culture was started. At this time, the distribution of the bacteria is uniform. Scale bar: 100 μm. **h** A photomicrograph of the fence-like channel. The color represents the concentration of the substance. Scale bar: 100 μm

hereafter referred to as mutualistic medium) was added to the reservoir. After 20 h, the communities generated by bacterial cultures containing only one strain did not grow, whereas the mixed-culture communities showed growth (see the "Methods" section). We also ruled out the possibility that bacterial strains that do not establish a functional mutually beneficial relationship would develop resistance in the chip (Supplementary Table 4). In the medium without CRO, the nutritional bacteria could grow alone, the resistant bacteria could not grow alone, and the two bacteria could grow together. Thus, in this medium, the two strains exhibit a parasitic relationship (Supplementary Table 5). Thus, it is clear that changes in the antibiotic pressure in the environment may change the relationship between the two strains from a parasitic

relationship to a mutually beneficial symbiosis, which is consistent with the results obtained by other researchers.

**An initial minimal population density is required for the growth of communities with obligatory mutualism.** In nutritionally deficient and antibiotic environments, the resistant and nutritional bacteria should theoretically grow mutually through functional collaboration. In our experiments, however, we observed that some communities did not grow (Fig. 2a). In addition, previous studies have shown that due to the Allee effect, a mutualistic community requires a minimal population density to grow[20,22]. Therefore, we reasoned that a low initial bacterial density may be responsible for the inability of these communities to develop.

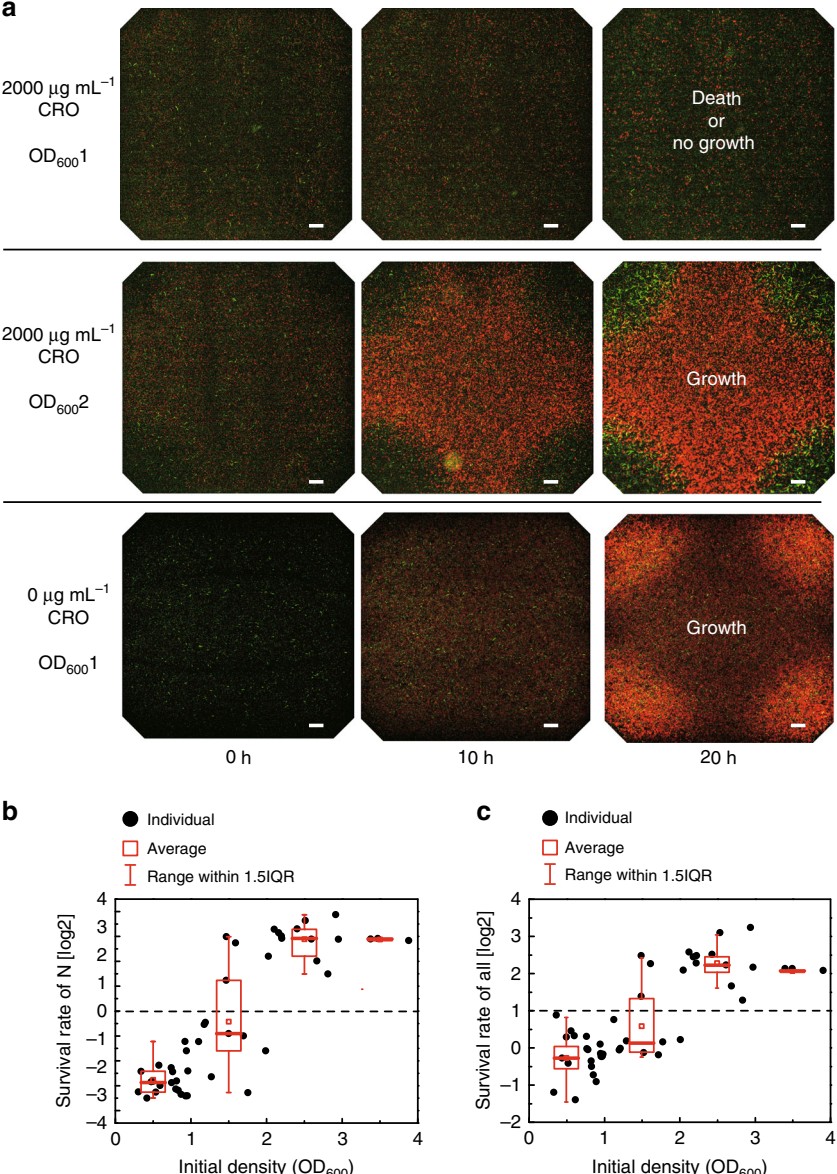

**Fig. 2** Minimal initial population density required for growth of the mutualistic community. **a** Two series of microscopic images of mutualistic communities with different initial identities cultured in mutualistic medium and one series of microscopic images of a parasitic community cultured in parasitic medium for 20 h (the GFP and RFP channels are superimposed). Scale bars: 100 μm. **b** Survival rates of the nutritional bacteria or **c** mutualistic community at different initial bacterial densities determined experimentally. Each black point represents the result from one sample. The superimposed box plot is drawn in red, and the x-axis boundaries of the groups occur at $OD_{600}$ values of 0–1, 1–2, 2–3 and 3–4. The number of biologically independent samples in each group is 20, 11, 12, and 3

To test this hypothesis and determine whether the growth of a mutualistic community requires a minimal population density, we conducted a series of experiments in which we changed the initial population density without changing the other experimental conditions and quantitatively analyzed bacterial population density by assessing the fluorescence intensity (Supplementary Fig. 3). To determine the growth status of the community at a certain time, we divided the number of bacteria at that time by the number of bacteria at time 0 h to calculate the survival rate (SR) at that time. As shown in Fig. 2b, c, under the given antibiotic concentration (2000 μg mL$^{-1}$), the SRs (calculated as the number of bacteria at 20 h divided by that observed at 0 h) of the nutritional bacteria or the community depended on the initial bacterial densities, and their qualitative relationships were similar to a high-order Hill function (or sigmoidal function), which separates two phases: the death

phase (SR of the nutritional bacteria ≪ 1) and the growth phase (SR of the nutritional bacteria ≫ 1).

Due to interferences from the experimental operation and image-processing errors, we could not determine the precise initial minimal population density required for community growth. Instead, we only determined a "critical density range" based on the statistical results: below the low density value, the probability for community growth is almost 0%, whereas above the high density value, the probability is almost 100%. Under the experimental conditions at which the data in Fig. 2 were obtained, the initial critical density range was identified as $OD_{600} = 1.0$–2.0. If the initial bacterial density is within the critical density range, the probability that the community can grow and the SR of the community is positively correlated to the density.

**Spatiotemporal dynamic behavior of the surviving communities**. An analysis of the spatiotemporal behavior of the survival communities revealed that the distributions of the communities became uneven after 6–10 h of growth, and after 20 h, this inhomogeneity became more obvious. In the environment without CRO, the two bacteria showing a parasitic relationship formed an uneven but colocalized distribution toward the nutrient source, whereas at high antibiotic concentrations, the bacteria in mutualistic communities exhibited an uneven and heterologous distribution (Fig. 2a). Additional images of the entire time series and samples of the two conditions are provided in the Supporting Information (Supplementary Figs. 4 and 5). Different interactions between populations results in different spatial distribution patterns of the communities[28]. In mutualistic communities, the resistant bacterial density showed a increase at the source side and the nutritional bacteria clustered in the center. This spatial distribution is worth further exploration. We think that the spatial distribution and the ratio of different bacteria may affect the growth rate of the community and may therefore change the optimal distribution according to environmental needs. To verify this hypothesis, the variation in the spatial distribution, the ratio of the two types of bacteria and the growth rate were carefully studied. Specifically, in this experiment, images of the observation area were obtained at 0, 2, 4, 6, 8, 10, 20, and 45 h after bacterial loading. To determine the distribution characteristics, we intercepted half of the diagonal region and divided it into 26 equal-width zones to calculate the growth rate in each zone (Fig. 3a). Figure 3b–d shows one typical example: for an initial bacterial density ($\sim OD_{600} = 2.5$) that was slightly higher than the critical density range, we plotted the SR distribution and the total growth rate as a function of time.

As shown in Fig. 3b, the community was almost evenly distributed at time 0 h. During the first 8 h, the total growth rate was negative, which indicated that the number of bacteria was lower than the initial number. During this period, obvious death of the nutritional bacteria was observed on the source side, and the resistant bacteria showed a gradual increase on the source side. After 8 h, the population growth rate became positive, and the distribution of the resistant and nutritional bacteria became different. The ratio of nutritional to resistant bacteria (the N/R ratio) was <1 during the first 8 h and increased distinctly to values notably >1 after 10 h, which indicates that the nutritional bacteria grow much faster than the resistant bacteria after 8 h. From 20 to 45 h, the community grew well and formed a relatively stable and heterologous distribution.

According to the results that the growth of a community is positively correlated to the initial density within the critical range, the growth rate should be lower during the period from 4 to 8 h than at the beginning. Notably, the population density at 4 h was lower than the initial density, indicating that this population is more likely to die out. However, during the period from 4 to 8 h, the total growth rate turned from negative to positive. An analysis of the SR of each zone in the diagonal region (Fig. 3c) revealed that the bacterial population distribution began to become nonuniform at ~4–8 h and that the total N/R ratio decreased to a value lower than the initial ratio. The results suggested that the changes in the spatial distribution and the N/R ratio of the community over time may result in an improvement in population growth.

**Verification of the roles of the community distribution and N/R ratio in the community SRs**. The data shown in Fig. 3b–d indicate that the critical density of a community is not the only indicator for evaluating the adaptability (death or growth in a given environment) of the community. The spatial distribution and N/R ratio of a community play an important role in its SR. To quantitatively study their effect, we employed a population mathematical model based on the reaction-diffusion equation to analyze the changes in the distribution and N/R ratio of the system and to simulate the critical densities and SRs under different conditions[4,16,17,20] (Supplementary Table 6).

The interactions between the variables in the model are shown in Fig. 4a. The nutritional (Y1) and resistant bacteria (Y2) could interact with lactose (N2) and CRO (A) that diffused from the external environment. The nutritional bacteria consume lactose to grow and secrete intermediate nutrients (N2), that are used by the resistant bacteria, while the resistant bacteria can degrade antibiotics. If the nutritional bacteria die due to the presence of antibiotics, they dissociate and release nutrients (N3, the value of which is notably greater than that of N2 at the source side in the beginning), which can also be used by the resistant bacteria. The parameters in the model were mainly derived from previous studies, but some parameters were fit from the "critical density" data. By conducting numerical simulations using this model, we determined the critical density and heterotopic distribution in the presence of a high antibiotic concentration, which qualitatively agree with the experimental results (Fig. 4d–f and Supplementary Fig. 6). We also verified that N3 was the key factor for the spatial structure, which explained why the resistant bacteria could grow rapidly near the source diffusion entrances (Supplementary Fig. 6).

To further verify the simulations and to test the influence of the impact of the distribution of the community on its growth, we designed a chip (Supplementary Fig. 7) that could artificially control the spatial distribution of bacteria and measured the SR of bacteria at three different spatial distributions with a relatively high bacterial density ($OD_{600} = 2.0–4.0$). The three distributions were as follows: uniform distribution, "natural" distribution, and "swap" distribution. For the natural distribution, the nutritional bacteria were only placed in the center, and the resistant bacteria were only placed near the source. The swap distribution involved swapping the natural distribution of the nutritional bacteria with that of the resistant bacteria. The experimental results are summarized in Fig. 5a. The consistent results obtained from the experiments and simulations confirmed that the SRs of the community varied with changes in the spatial distribution and that the SR of the natural distribution was greater than that of the uniform distribution (Fig. 5a, Supplementary Fig. 8). We thus changed the N/R ratio while maintaining the same total number of bacteria in the initial state and found variations in the SRs of the communities, as shown in Fig. 5b and Supplementary Fig. 9. For an initial bacterial density of $\sim OD_{600} = 2.0–4.0$, the SR of a community with an N/R ratio of 1:3 was greater than that of communities with N/R ratios of 1:1 and 3:1, and this finding was obtained in both the simulations and experiments.

Furthermore, we attempted to construct the SR landscape under different conditions (different initial distributions, different initial N/R ratios, and different initial bacterial densities). First, we kept the initial N/R ratio constant (N/R = 1), simplified the distributions of some experimental traces, calculated the critical number, which indicates the drug resistance ability, of these distributions and arranged the distributions based on the critical number, as shown in the left of Fig. 5c. A smaller critical number indicated that the corresponding distribution has a higher drug resistance pattern. Using different initial bacterial densities and distributions (ordered based on the critical number), we then calculated the SR of the N bacteria and the community (time period of 3 h) and drew the heat maps shown in Fig. 5c and Supplementary Fig. 10. The line corresponding to a SR of 1 in the heat map has a negative slope, which shows that the upper distributions contribute to a growth advantage to the lower

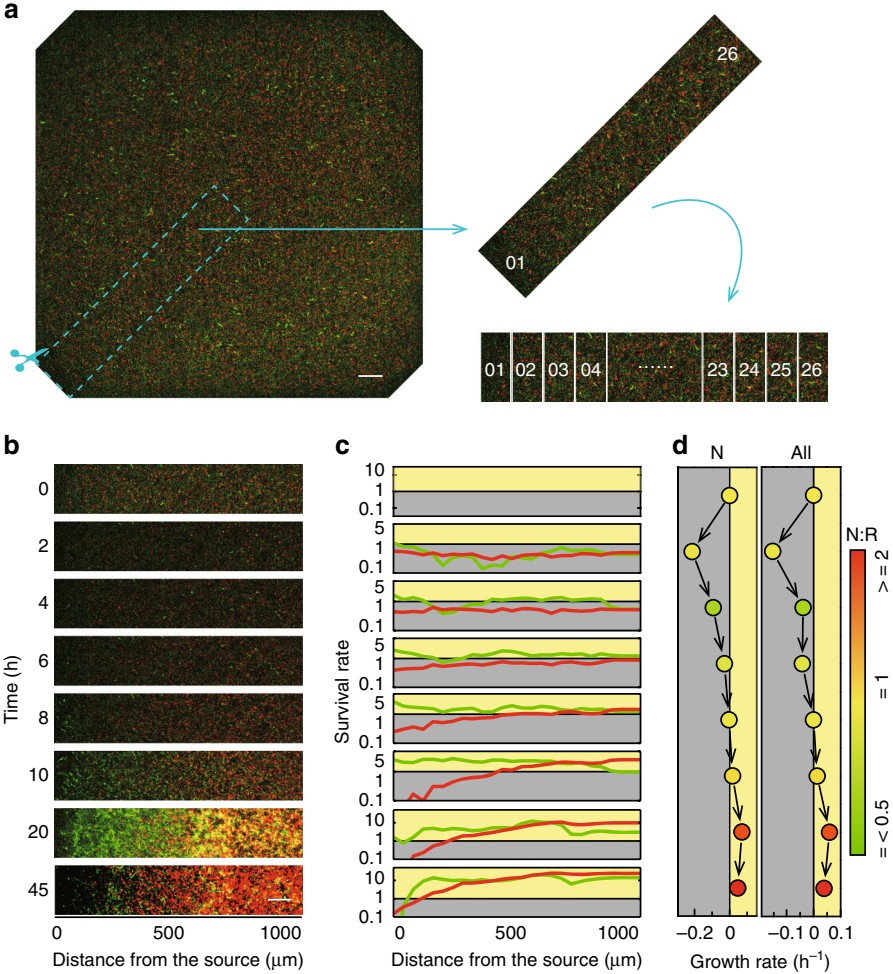

**Fig. 3** Changes in the growth of the community over time and space determined experimentally. **a** Intersection and division of the diagonal regions of the micrographs. The dashed-line box indicates the diagonal region that was intercepted for analysis. The area was divided into 26 small equidistant regions. Among these, region No. 1 is closest to the source, while region No. 26 is closest to the center. Scale bar: 100 μm. **b** Time-lapse micrograph of the diagonal region. **c** Spatial distribution of the survival rate of the two strains as a function of time. The green and red lines show the survival rates of the resistant and nutritional bacteria, respectively. Scale bar: 100 μm. **d** The growth rates of the nutritional bacteria and the total community as a function of time. The color of each dot was derived from the N/R ratio at the time point

distributions around the critical density. We also plotted the heat maps for initial N/R ratios of 3/1 and 1/3. The SR line moved toward a higher (lower) bacterial density as the N/R ratio increased (decreased).

**Changes in bacterial distribution and N/R ratio enhance community SR**. We subsequently traced the temporal trajectories of the changes in the community spatial distribution, N/R ratio and quantity for different initial densities through experiments (Fig. 6) and simulations (Supplementary Fig. 11). In experiments, by analyzing the representative quantity change trajectories in each density range, three typical samples were obtained (Supplementary Figs. 12 and 13). The microscopic images of the three samples were processed to obtain the nutritional bacterial or community densities, the N/R ratio and the simplified distribution characteristics at different times (Supplementary Fig. 14). The experimental trajectories were then also plotted for further analysis (Fig. 6).

As shown in Fig. 6, when the number of bacteria decreased such as samples 1 and 2, the distribution changed to upper distributions with a smaller "critical density" and the N/R ratio decreased at the same time. These results indicated that the

community has a much higher SR compared with that of a community with a uniform distribution and an N/R ratio of 1, as indicated in Fig. 5c. Different fates were found for sample 1 (death) and sample 2 (growth). For an initial number smaller than the "critical number", the distribution changed to the top distribution, and the N/R ratio changed to a very small value but did not reach the "turning" point (defined as the time point when the number of bacteria present changes from decreasing to increasing). The trajectory of sample 2 showed a clear turning point. The number of bacteria of the community, which was smaller than the initial number, began to increase after evolving to an upper distribution and a lower N/R ratio, even when the number is smaller than the initial number. Whereas at the initial state, the number of bacteria decreased. After the number of bacteria in the communities of samples 2 and 3 increased, the trajectories changed toward the right-lower area, and the N/R ratio increased, indicating that the right-lower area of Fig. 5c represents the attractor area of all growth samples.

**Mutualism can use spatial freedom to promote survival under antibiotic stress**. Because mutualistic communities can use available space to obtain a specific distribution and thus improve

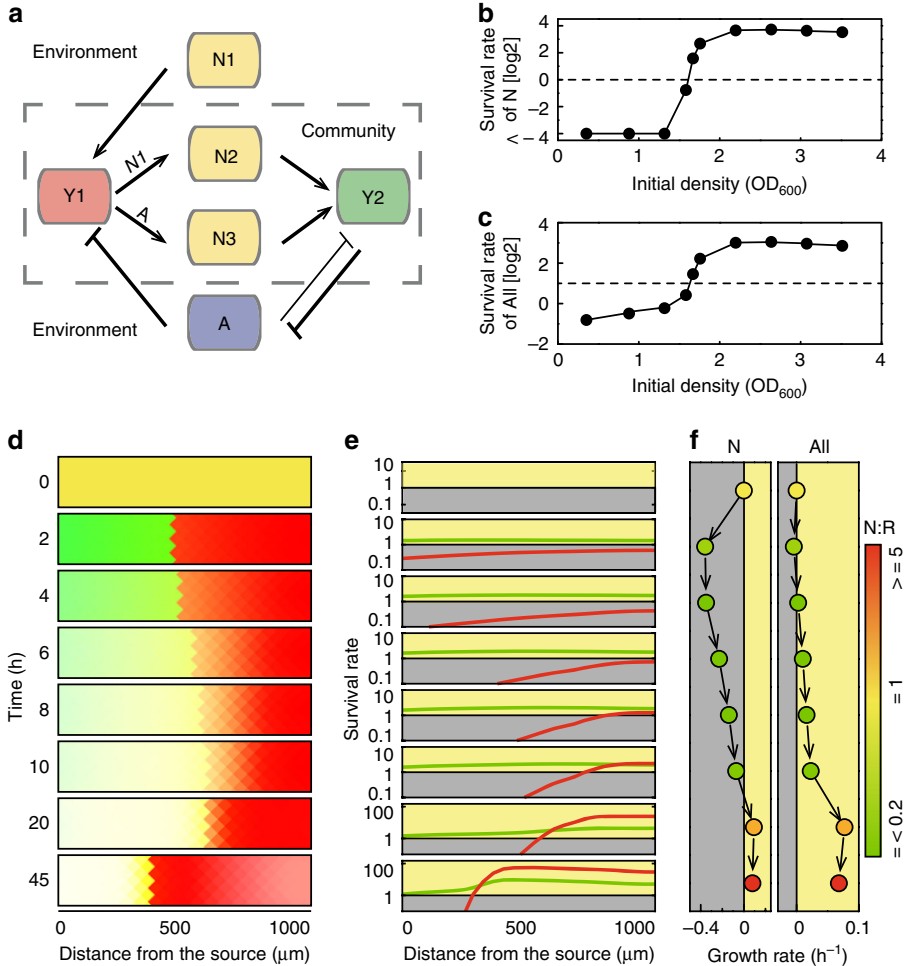

**Fig. 4** Model and simulation results of the bacterial community. **a** Interaction of two bacterial strains and the surrounding chemicals. The dashed line shows the boundary between the community and the environment. **b** Simulation results of the survival rates of the nutritional bacteria or **c** the mutualistic community with different initial bacterial densities. **d–f** Temporal simulation results of the spatial distribution, survival rate distribution, and growth rates of the nutritional bacteria and the total bacterial community. The colors of the dots shown in **d** were derived from the N/R ratio at the time points

their SR, we considered the different spatial scales that limit the growth of communities in the natural environment[16,29] and conducted simulations and experiments to explore the effect of growth space restriction on community growth.

In this study, we used two simulations: in one simulation, the chemicals were maintained at uniform levels (rapid mixing in the culture area), and the other simulation reflected the experimental conditions. In the first simulation, the critical numbers (critical density plus system volumes) remain almost constant with different culture spaces. In contrast, in the simulation in which the bacteria and chemicals show varying distributions, the critical numbers (the indicator of community resistance) decreased with increase in the size of the culture area (Supplementary Fig. 15).

To verify the spatial effect, we used chips with culture regions of different sizes (400 or 800 μm in length) for our experiments (Supplementary Fig. 16). While keeping the fence size unchanged, we changed the culture areas by changing the spaces and used a series of initial densities to find the critical density for different spaces. The results showed that the critical density range of the mutualistic community in the 800-μm chip was $OD_{600} = 8–16$ (Fig. 7), which indicated that a two-fold higher community density was required for growth in the 800-μm chip compared with that required for growth in the 1600-μm chip. In the 400-μm chips, even though the density was very high, no community growth was observed, which may be due to the fact that the

density required for survival has not been reached. The results obtained in the experiments were consistent with the simulation results, and both the simulation and the experimental results suggested that the community can utilize its spatial freedom to promote their survival under antibiotic stress.

## Discussion

We observed that an initial $OD_{600} = 1–2$ was needed for growth in the 1600-μm culture system, whereas a much higher initial bacterial density (such as an $OD_{600} = 8–16$ for the 800-μm culture system) was needed for a smaller culture system to resist the antibiotics present in the medium. These densities are much higher compared with those observed in minimum inhibitory concentration (MIC) studies[25] using a similar microfluidic system and is smaller than that of formed biofilms in other microfluidic systems ($\sim 3 \times 10^9–10^{10}$ cells mL$^{-1}$)[15,30]. By analyzing the changes in the distribution and bacterial ratio of artificially constructed mutually beneficial mutualistic systems that grow freely in a finite space, we found that the distribution and ratio of bacteria in the community affect the drug resistance of the community. Furthermore, under antibiotic pressure, the bacterial population could spontaneously adjust its spatial distribution and bacterial ratio to increase its drug resistance. Our results complement current knowledge regarding the relationship between

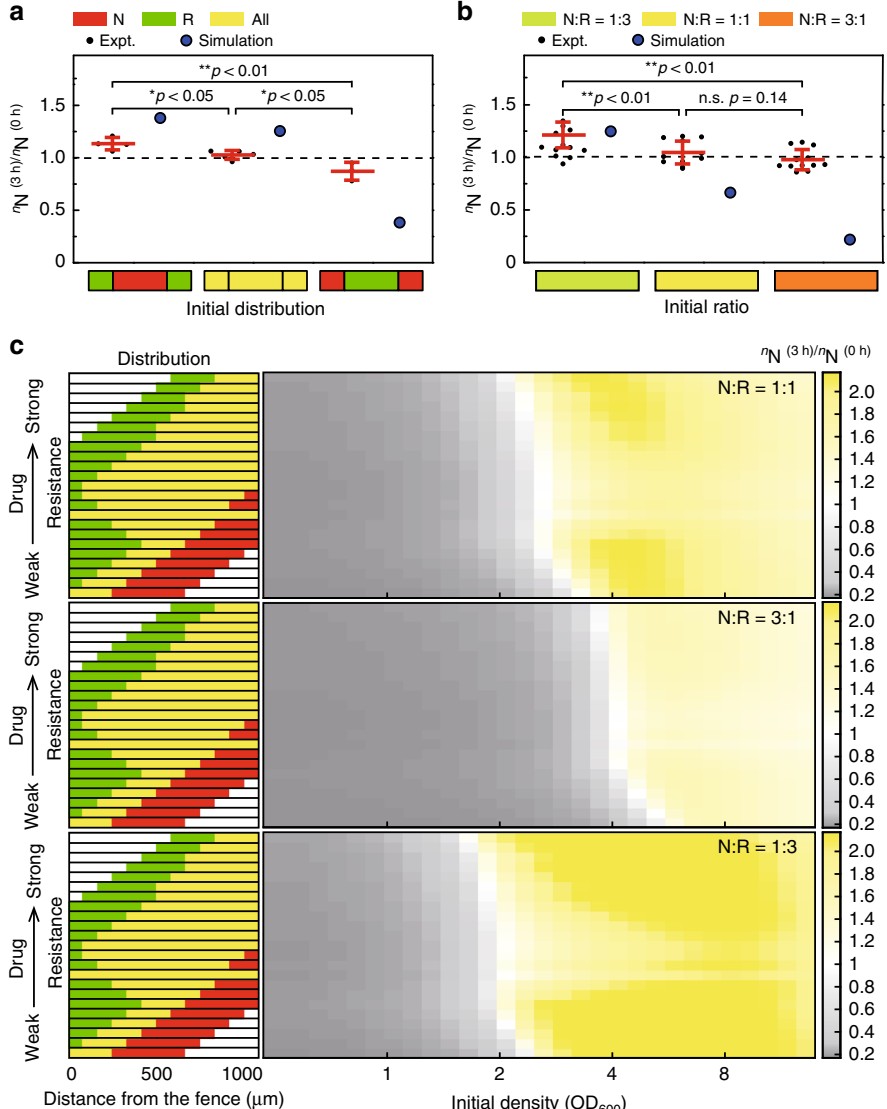

**Fig. 5 Influence of the distribution or N/R ratio on the bacterial survival rate. a, b** Communities exhibit different survival rates based on their initial distributions or N/R ratios. The three rectangles drawn below the histogram represent the three distributions or N/R ratios. The ordinate shows the survival rate of the nutritional bacteria at 3 h (SI, Figs. S8 and 9). Error bars represent standard error of the mean. The asterisk indicates the significance of the difference between the samples in each group. "*" indicates that the P-value of the t-test is <0.05, and "**" indicates that the P-value of the t-test is <0.01. There are more than three replicate samples in each group: natural distribution, $n = 4$; uniform distribution, $n = 5$; sawp distribution, $n = 3$; different N/R ratios, $n = 12$. **c** Maps of the nutritional bacterial survival rate under different conditions (different initial distributions, different initial N/R ratios, and different initial ODs) obtained by the simulations

spontaneously formed community spatial structures and environmental adaptability. The spatial structures observed in our experiments are different from those obtained in previous models or experiments[28,31]. This difference may be caused by the closure of the community growth space and the complexity of interactions between the populations. Moreover, our findings can provide guidance for artificially interfering with the formation of spatial structures, which would aid the treatment of pathogenic biofilms and the accumulation of microorganisms in wastewater treatment facilities[15,32].

Due to the higher growth and mutation rates of microorganisms, microbial communities are likely to change functions beyond those designed by the system[16,33]. Although the influence of evolution is not discussed in this paper, it is possible to explore the evolutionary behavior of complex communities using our experimental system. In addition, in nature, the relationships or

functions of bacteria are far more complex than those that occur in our system, such as the function of chemotaxis, which can change the community spatial distribution at a much faster rate than that obtained due to growth or death. The introduction of chemotaxis into this system may make it possible to accelerate the formation of a tolerant distribution, which is also a question worth investigating further.

## Methods

**Strains, plasmids, and materials.** The resistant strain used in this study was *E. coli* DH5α transformed with the pUA66-GFP plasmid, which harbors a gene encoding extended-spectrum β-lactamase (CTX-M) and a gene encoding GFP. The nutritional strain was *E. coli* MG1655 transformed with a plasmid harboring a gene encoding RFP. More Information regarding the genotypes, phenotypes, and sources of the bacterial strains and plasmids used in this study are listed in Supplementary Table 1. Tryptone and yeast extract were purchased from Oxoid, Ltd. Agar was purchased from Sigma, and NaCl and lactose were obtained from

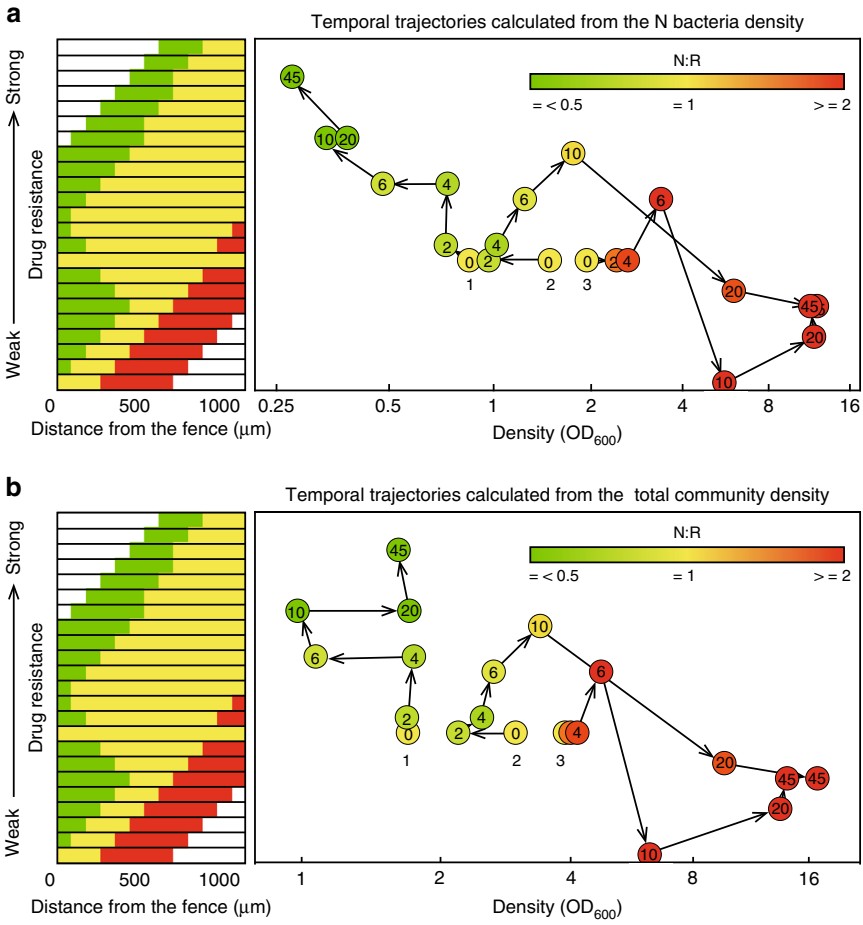

**Fig. 6** Temporal trajectories of samples with different initial densities determined experimentally. The dots and arrows illustrate the temporal trajectories calculated from (**a**) the nutritional bacteria density or (**b**) the total community density. The numbers in the circles indicate the experimental time points (*h*) and the numbers below the circles indicate the number of the different exemplary samples. The color of the dots indicates the N/R ratio

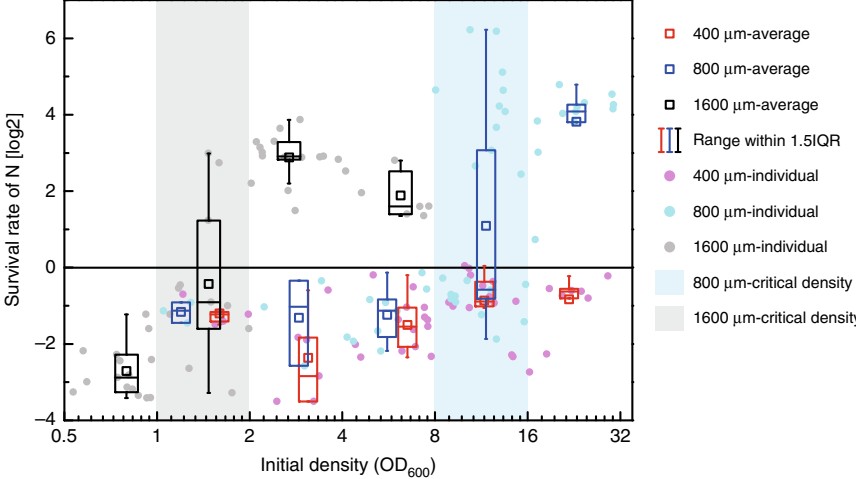

**Fig. 7** Experimental survival rates of communities in different culture areas. The dots with different colors represent the average survival rates of the nutritional bacteria in chips with different culture area side lengths (400, 800, and 1600 μm) after 20 h of culturing. The *x*-axis boundaries of the superimposed box occur at $OD_{600}$ values of 0.5–1, 1–2, 2–4, 4–8, 8–16, and 16–32. The number of biologically independent samples in each group is 0, 5, 6, 14, 13, 8 (in the 400-μm chips), 0, 3, 3, 9, 22, 12 (in the 800-μm chips), 15, 11, 15, 7, 0, 0 (in the 1600-μm chips). The *x*-coordinate of each box is the average of the *x*-coordinate of the samples it contains. The grayish and light blue rectangular areas mark the critical density range of the communities grown in the 1600 and 800 μm chips

Sinopharm Chemical Reagent Co., Ltd. PBS was purchased from Shanghai Double-Helix Biotech Co., Ltd., and M9 was obtained from Leagene. CRO and kanamycin were purchased from Rocephin, and ampicillin sodium salt was purchased from Amresco.

**Validation of the strain characteristics**. To verify the tolerance to CRO and the utilization of lactose, each strain was inoculated individually into 2 mL of LB medium containing 2 μg mL$^{-1}$ CRO or M9 medium containing 0.8% lactose in tubes. The cultures were shaken at 220 rpm at a temperature of 37 °C. The results are shown in Supplementary Table 2.

**Verification of the mutually beneficial mutualism of the communities in tubes**. Each strain of the bacteria was cultured in the appropriate culture medium (Supplementary Table 3) overnight for 15 h (at 37 °C, 220 rpm), and the culture was diluted 20-fold into fresh medium for 2 h. The 2-mL culture solution was then centrifuged at 500 rcf and resuspended in 1 mL of PBS. The bacterial density was measured using a spectrophotometer (at 600 nm), and four types of bacterial strains were mixed in equal quantities and added to different wells of 200 μL of medium (M9 containing 100 μg mL$^{-1}$ CRO, 250 μg mL$^{-1}$ Amp and 0.8% lactose) in 96-well plates. The plates were then placed in a microplate reader and incubated at 37 °C and 600 rpm. The OD$_{600}$, GFP fluorescent and RFP fluorescent of the mixed culture were measured by the microplate reader at each hour. The results are shown in Supplementary Fig. 1. Each set of data was obtained from seven to eight repeated experiments. Only the combination of resistant bacteria with nutritional bacteria showed obvious growth. The results ruled out the possibility that the mutualism obtained with resistant and nutritional bacteria could be mutualistic through functions other than the two mutually beneficial interaction functions of degrading antibiotics and providing intermediate nutrients.

**Design, processing, and use of chips**. After preparing the chips according to standard soft-lithography methods[26], we filled the fence with a low-melting-point agarose gel to form a semipermeable membrane. Prior to the bacterial culture experiments, PBS buffer was added to the reservoir and the culture area to prevent the gel from drying out.

The PBS suspension of the required bacteria was prepared using the above-mentioned method. According to the experimental design, the suspension–solution mixture of the two types of bacteria was added to the chip in proportion to the cell inlet. After the bacteria were distributed evenly in the culture area, the residual suspension solution was aspirated out, and the gel was then added to seal the inlet channel. The PBS buffer in the reservoir was then replaced with the specific medium required for the experiment. To maintain the bacteria in a nutritious environment, 100 μL of medium was added to each reservoir. Wetted cotton balls were added to the plate in which the chips were placed to avoid volatilization of the medium. The plates were then cultured in an incubator at 37 °C.

The medium could be changed at any time according to the experimental design during the cultivation. Considering the degradation of antibiotics and the potential consumption of nutrients, we renewed the medium in the reservoir every 10 h to ensure that the antibiotics and nutrients at the diffusion point were nearly stable throughout the culture.

**Determining the diffusion time scale of lactose and CRO in the chip**. First, we used MATLAB to simulate the diffusion of lactose from the reservoir to the culture area in the chip. The diffusion coefficient of lactose in water is $4.3 \times 10^{-10}$ m$^2$ s$^{-1}$. Prior to diffusion, the concentration of lactose in the culture area was 0, and during the diffusion process, the concentration of lactose in the fence linked to the reservoir was maintained at 0.023 mol L$^{-1}$ (the mass fraction was 0.8%). The results are shown in Fig. S2a. Approximately 100 min later, the lactose was almost uniformly distributed. The diffusion coefficient of CRO is similar to that of lactose; thus, the diffusion results were similar.

The diffusion coefficient ($5 \times 10^{-10}$ m$^2$ s$^{-1}$) of fluorescein isothiocyanate (FITC) is similar to those of lactose and CRO, and its diffusion can be observed under a fluorescence microscope. Therefore, we used FITC to verify the diffusion rate in the chip. We added 1 mM FITC to the reservoir and then used a fluorescence microscope (Nikon Eclipse Ti-E) to obtain images every 20 min. An exposure time of 300 ms was used. The environmental conditions used for the diffusion testing (37 °C, no shaking and the use of moisturized cotton balls) were similar to the final conditions used in the microfluidic chip. As shown in Supplementary Fig. 2, fluorescein can diffuse evenly within 2 h, which is consistent with the results obtained with MATLAB.

**Photomicrograph conditions**. To observe and record the in situ growth of bacteria, we photographed the culture area using a Nikon Ti-E fluorescence microscope and Andor iXon 897 camera. The shooting time was selected based on the experimental design. During the shooting, to capture the whole culture area and improve the definition of the photograph as much as possible, we chose different magnifications according to the size of the culture area: the chips with 800 and 1600 μm side lengths in the culture area were amplified by a ×4 objective lens, and the chips with a 400-μm side length were amplified by a ×10 objective lens. To record the boundary of the culture area and the growth state of the bacteria labeled

with GFP or RFP simultaneously, we selected three channels, namely, bright, GFP, and mCherry channels, for shooting. For each image, the boundaries of the culture area were determined using the bright channels, and the culture area was then fixed in the same position for subsequent imaging.

**Verification of the mutually beneficial mutualism of communities in the chips**. After verifying the mutualistic relationship of the communities in tubes using the controlled experiments, we repeated the same tests in the prepared chips to confirm that the communities would have the same characteristics and relationships when cultured in a chip. The density of the culture was OD$_{600}$ = 2.0. The methods used for the chip experiment were described above. The medium used for the coculture of the two strains was M9 medium containing 250 μg mL$^{-1}$ Amp, 2000 μg mL$^{-1}$ CRO, and 0.8% lactose.

In the coculture samples, only the coculture containing the resistant and nutritional bacteria exhibited significant growth after 20 h of culture (Supplementary Table 4). The above results agree with those obtained from the tube experiments.

**Change in the relationship of the community under different culture conditions**. When cultured in chips, the nutritional bacteria could grow on their own in the medium without CRO and with lactose as the only carbon source (M9 medium containing 0.8% lactose and 250 μg mL$^{-1}$ Amp), but the resistant bacteria could not. When the two strains were mixed in this medium, the resistant bacteria could rely on the nutrients released by the nutritional bacteria to grow; thus, under this condition, the two strains showed a parasitic relationship (Supplementary Table 5).

**Extraction of quantitative data from microscopic images**. First, we selected the culture area from the image for analysis. Because the position of the culture area is fixed for each image, we were able to use MATLAB to evaluate the information in the same area. To determine the number of each type of bacteria in the culture area, we decided to use the fluorescence intensity of the microphotographs because each type of bacteria is labeled with a fluorescent protein, and the fluorescence intensity of multiple bacteria is normally distributed. Therefore, within a certain bacterial number range, the average fluorescence intensity of a region is linearly correlated with the number of bacteria in the region. To distinguish the background fluorescence from the bacterial fluorescence, the distribution of the fluorescence intensity of all the pixels in the culture area was counted, and the distribution curve of the background fluorescence was then obtained. The average fluorescence intensity of the remaining pixels was calculated after removing the background fluorescence according to the distribution curve.

To determine the accuracy of the above-described quantitative method, we first verified the linear relationship between the average fluorescence intensity and the number of bacteria. We added the bacterial suspensions with different densities evenly to the closed observation area and obtained multiple photographs of each sample. The fluorescence intensity of each photo was read according to the above-described method, and the average value from multiple samples of each density was then obtained. The results (Supplementary Fig. 3) showed that the OD$_{600}$ of the bacteria labeled with RFP or GFP exhibited a good linear correlation with the corresponding average fluorescence intensity ($R^2 = 0.95$) within a certain density range (OD$_{600}$ = 0.25–32). To determine the confidence range of the data, we obtained a number of blank control photographs without bacteria and found that their red or green fluorescence intensity was substantially lower than 0.1. The linear relationship between the bacterial density and average fluorescence intensity was used in the subsequent analysis.

**Quantitative analysis of growth status**. The SR at a certain moment and position can be calculated as

$$SR(x, t) = i(x,t)/i(x, 0)$$

where $i(x, t)$ and $i(x, 0)$ are the GFP or RFP fluorescence intensities within position $x$ at time $t$ and 0 h, respectively.

The total community growth rate (GR) at a specific moment can also be calculated as

$$GR(t) = \frac{\ln[(I_R(t) + I_N(t))/(I_R(0) + I_N(0))]}{t}$$

where $I_R(t)$, $I_N(t)$, $I_R(0)$, and $I_N(0)$ represent the total GFP (the resistant bacteria, R) and RFP (the nutritional bacteria, N) fluorescence intensities at time $t$ and time 0 h.

We also calculated the total growth rate of the nutritional bacteria to show a clearer trace of growth compared with that of the total community:

$$NGR(t) = \frac{\ln[I_N(t)/I_N(0)]}{t}.$$ The ratio of the two types of bacteria was defined as the N/R ratio (the total number of the nutritional bacteria to that of the R bacteria).

**The growth state of the community various under different culture conditions**. The resistant and nutritional bacteria could grow together in the medium containing lactose as the only carbon source and no CRO (M9 medium containing 0.8% lactose and 250 μg mL$^{-1}$ Amp), which indicated a parasitic relationship between the bacteria. After this parasitic community was cultured in a chip for 20 h, the densities of both types of bacteria were significantly increased at the corners.

The two species of bacteria formed an uneven and colocalized distribution (Fig. 2a and Supplementary Fig. 5).

**Quantitative extraction of distribution characteristics in the diagonal region**. Because the chip was spatially symmetrical in the center and the distribution of the community was also centrosymmetric, we could express the whole distribution based on the distribution at the corner of 1/4 of the total space size. In addition, the community formed an arc-shaped distribution in this region. Similar numbers of bacteria were obtained at similar distances from the nutrition and antibiotic source. We further selected the distribution of the diagonal region to represent the spatial distribution of the entire observation area. As a result, we abstracted the two-dimensional continuous distribution into a one-dimensional discrete distribution.

In order to quantitatively describe the spatial distribution characteristics of each strain in the community at any given time, we first intercepted the diagonal region from the fence to the center and then divided the region into several strips of the same size perpendicularly to the diagonal direction, as shown in Fig. 3a. The number of bacteria in each strip was read. Each sample had four such diagonal regions, and after the numbers for one region were separately obtained, we calculated the average from the four sets of data. The distribution characteristics of the bacteria obtained using this method are consistent with the observed distribution characteristics.

**Simulation model**. To explore the related factors affecting the community's resistance characteristics and to predict the resistance behavior of the community, a mathematical model based on the reaction diffusion equation was established.

A grid of 40 μm × 40 μm was used as the minimum space unit, and thus, the culture area of a chip with a side length of 1600 μm can be abstracted into a grid of 40 × 40. In the calculation, we used a 40 × 40 matrix to correspond to the above grid, and the position information (x, y) for each variable can thus be provided. Although the reactions that occurred in the experiments are all continuous events, time was also discretized for ease of calculation. The time interval of each calculation round was 0.1 s. The main variables in the model were as follows: the number of nutritional bacteria (Y1) and resistant bacteria (Y2) at position (x, y) and time t, $y_1(x, y, t)$ and $y_2(x, y, t)$; the concentration of lactose (N1), $n_1(x, y, t)$; the concentration of nutrients secreted by nutritional bacteria that can be used by resistant bacteria (N2), $n_2(x, y, t)$; the concentration of nutrients released by bacterial death (N3), $n_3(x, y, t)$; and the concentration of CRO (A), $a(x, y, t)$. The interactions among the variables can be briefly described as follows: N1 is consumed only by Y1 and produces N2, A is only degraded by Y2, and both Y1 and Y2 can consume N2 and N3 and are suppressed by A.

In each grid, the calculation relationships between variables in each calculation interval were defined as follows:

$$\frac{\partial a}{\partial t} = D_A \nabla^2 a - \frac{c_3 a}{C_3 + a} y_2 \cdot \left( \frac{N_{max}}{y_1 + y_2 + N_{max}} \right) \tag{1}$$

$$\frac{\partial n_1}{\partial t} = D_{N_1} \nabla^2 n_1 - \frac{c_1 n_1}{C_1 + n_1 + n_2 + n_3} y_1 \cdot \left( \frac{N_{max}}{y_1 + y_2 + N_{max}} \right) \tag{2}$$

$$\frac{\partial n_2}{\partial t} = D_{N_2} \nabla^2 n_2 + \left( \lambda_{N_2} \cdot \frac{c_1 n_1}{C_1 + n_1 + n_2 + n_3} y_1 - \frac{c_1 n_2}{C_1 + n_1 + n_2 + n_3} y_1 - \frac{c_2 n_2}{C_2 + n_2 + n_3} y_2 \right) \cdot \left( \frac{N_{max}}{y_1 + y_2 + N_{max}} \right) \tag{3}$$

$$\frac{\partial n_3}{\partial t} = D_{N_3} \nabla^2 n_3 + \left[ \lambda_{N_3} \cdot R \cdot \left( \frac{k_2 a}{K_2 \cdot \frac{n_1 + n_2 + n_3 + 100}{n_1 + n_2 + n_3 + 1} + a} y_1 + \frac{k_5 a}{K_5 \cdot \frac{n_2 + n_3 + 1}{n_2 + n_3 + 0.01} + a} y_2 \right) - \frac{c_1 n_3}{C_1 + n_1 + n_2 + n_3} y_1 - \frac{c_3 n_3}{C_2 + n_2 + n_3} y_2 \right] \cdot \left( \frac{N_{max}}{y_1 + y_2 + N_{max}} \right) \tag{4}$$

$$\frac{\partial y_1}{\partial t} = \left( \frac{k_1 (n_1 + n_2 + n_3)}{K_1 + n_1 + n_2 + n_3} - \frac{k_2 a}{K_2 \cdot \frac{n_1 + n_2 + n_3 + 100}{n_1 + n_2 + n_3 + 1} + a} \right) \cdot y_1 \cdot \left( \frac{N_{max}}{y_1 + y_2 + N_{max}} \right) \tag{5}$$

$$\frac{\partial y_2}{\partial t} = \left( \frac{k_4 (n_2 + n_3)}{K_4 + n_2 + n_3} - \frac{k_5 a}{K_5 \cdot \frac{n_2 + n_3 + 1}{n_2 + n_3 + 0.01} + a} \right) \cdot y_2 \cdot \left( \frac{N_{max}}{y_1 + y_2 + N_{max}} \right) \tag{6}$$

Each formula can be described as listed below:

Eq. (1) indicates the change in antibiotic (A), which can only be decomposed by the resistant bacteria (Y2).

Eq. (2) indicates the change in lactose (N1), which can only be consumed by nutritional bacteria (Y1).

Eq. (3) indicates the change in the intermediate product (N2), which is the intermediate product secreted with a conversion rate of $\lambda_1$ and can be consumed by Y1 and Y2.

Eq. (4) indicates the change in N3. If the nutritional bacteria (Y1) or the resistant bacteria (Y2) die, nutrition (N3) will be released by rupture at a conversion ratio ($\lambda_{N3}$). N3 is utilized by Y1 and Y2, and the dead bacteria will gradually release N3 with a decay time of 1000 s.

Eq. (5) indicates the change in the nutritional bacteria (Y1). The growth and death of Y1 are related to N1, N2, N3 and A, and the chemoattractant for N1 is ignored in this equation.

Eq. (6) indicates the change in the resistant bacteria (Y2), which grows by using N2 and N3 and can degrade the antibiotic A.

If the nutritional level is very low, the drug resistance of the bacteria will be increased, and we thus set a nutritional condition for K2 and K5.

All the parameters are listed in Supplementary Table 6.

The above equations include the limitation of the finite space of the bacteria and the diffusion of nutrients and antibiotics between the grids. The strains used in the experiment were weak in terms of exercise capacity, and their positions did not change substantially in the absence of external force. The growth of bacteria will push the surrounding bacteria, which might change the position of the bacteria. If the density of the nascent bacteria is higher than that of the surroundings, a higher density increases the likelihood that the bacteria at that location will be pushed to surrounding locations. Therefore, in the model, we have added a simulation of this push effect.

To verify the accuracy and predictability of the model, we designed a series of verification simulations. First, we simulated various extreme conditions, including the following: we cultured each strain by itself; we did not add lactose; we used nutritional bacteria that could not produce intermediate nutrients; and we generated conditions in which bacterial death did not release nutrients. The simulation results were consistent with the experimental or expected results. We subsequently selected representative experimental results and performed simulations using the experimental conditions. The simulation results were in good agreement with the experimental results, as shown in Figs. 3 and 4. The medium used in the experiment was M9 containing 0.8% lactose, 250 μg mL$^{-1}$ Amp and 2000 μg mL$^{-1}$ CRO. The initial density of the bacteria was ~OD$_{600}$ = 2.5, and the incubation time was 20 h. In the simulation, the corresponding parameters for each substance in the medium were as follows: $n_1$ was 100, $a$ was 2000, the initial amount of $y_1$ and $y_2$ was 25,000 (~OD$_{600}$ = 2.2), and the simulation time was 20 h. In the simulation image (Supplementary Fig. 6), the red intensity indicates the density of nutritional bacteria, the green intensity indicates the density of resistant bacteria, and the yellow color represents the overlap of the two bacteria. The simulation results also revealed a heterotrophic distribution in which the resistant bacteria are near the source side and the nutritional bacteria are close to the center.

The above simulations were all performed using MATLAB.

**Calculation of critical density by using the simulation**. In the experiment, we found a critical density for growth of the mutualistic community. In other words, in the presence of a constant antibiotic concentration, a mutualistic community can effectively increase if the initial density is greater the critical density. We used the above-described model to perform a series of simulations and thus determine whether the same phenomenon could be simulated.

For immobilized N$_1$ and A, the initial numbers of Y$_1$ and Y$_2$ were changed in the range of 1000–100,000 to simulate the growth state of the community after 20 h. The results were consistent with the experimental findings. The growth rate of the community was positively correlated with the initial bacterial density, and a critical density was observed. Therefore, in the simulations, we defined the critical density of the distribution as the minimum initial density at which the community could grow in the presence of specific antibiotics and nutrient concentrations. Figure 4 shows the relationship between the SR and the initial density of the mutualism obtained from the simulation of a culture grown in a 1600-μm chip with a CRO concentration of 2000 μg mL$^{-1}$. Under this condition, the critical density of mutualism was found to be ~OD$_{600}$ = 1.6 which is consisted with the experimental results.

**Design and application of the stripe chip to control initial spatial distribution**. To verify the difference in community resistance caused by different initial distributions through experimentation, we designed a stripe chip that can obtain the three above-mentioned initial distributions (Supplementary Fig. 7a). The culture area of the chip is a rectangle with a length of 1100 μm, and the length is approximately half of the diagonal of the 1600-μm square chip. Only one end of the culture area is connected to the reservoir through a gel fence, corresponding to the corner of the square chip. The other end is closed, corresponding to the center of the square chip. Therefore, the distribution of bacteria in the culture area is a one-dimensional simplification of the bacterial distribution in the 1600-μm square culture area, and the properties are similar. The two bacterial passages divide the culture into two regions, A and B. Different initial bacterial distributions can be arranged by adding different bacteria to each region through different channels.

A uniform distribution could be obtained by adding both resistant and nutritional bacteria at a 1:1 ratio to regions A and B. The addition of the nutritional bacteria to region A and of the resistant bacteria to region B could yield the natural distribution formed by the evenly distributed community after a period of growth. Changing the positions of the two strains from the above-described distribution yielded the opposite distribution. In the experiments, the ideal initial distributions were obtained using the chip (Supplementary Fig. 7b).

**Calculation of the critical density of different distributions using the model**. Similar to the experimental images, the diagonal region of the simulation matrix could also abstract the distribution of the entire culture area into a one-dimensional discrete distribution. Moreover, because the distribution of the bacteria present in the experiment was continuous, we only considered the case in which each strain had only one continuous distribution area (band) on the diagonal. We further assumed that each of the bacteria showed a uniform distribution area, i.e., there was no quantitative difference between each grid in the distribution area. Based on these two hypotheses, we extracted two parameters from the one-

dimensional distribution of each strain to fully represent the spatial distribution of this strain. The parameters were the distance ($d$) from the edge of the bacterial band from the source side and the width of the bacterial band region (delta). Different spatial distributions of each strain could be constructed by adjusting these two parameters, and the distributions of the community could be obtained by combining the spatial distribution of the two strains.

The growth rates of communities with different initial distributions and the same number of bacteria were then simulated (Supplementary Fig. 8). Additionally, using the above-described method, the critical densities of different initial distributions were obtained through simulations.

### Analysis of typical trends of mutualism during culture in a chip.
The trend for the change in the community number can be obtained from an analysis of time-series data of the numbers of each strain in one sample. The growth rate can be calculated using the formula mentioned in the main text.

Although the trend in the number varied among samples, the variation of the trajectory was related to the initial density. Most samples with a low initial density died. With increases in the initial density, the proportion of the samples that grew after an initial death (abbreviated as "turn") first increased and then decreased. The proportion of the samples that grew continuously increased gradually (Supplementary Fig. 12).

Therefore, we determined the typical trends of the samples with each density range as follows. If the initial density was lower than $OD_{600} = 1$, the community died. If the initial density was in the medium range ($OD_{600} = 2–3$), the community first died and then grew. At higher initial densities ($OD_{600} = 3–6$), the community continually grew. Thus, one sample from each of these groups of samples was selected as the typical sample. The specific results are shown in Supplementary Fig. 13.

### Extraction of the community distribution from experiments and simulations.
The distribution of a community affects its drug resistance; therefore, in addition to the changes in the number of bacteria, the time-series trends of the community distribution are also very important. To analyze the distribution trend, the distribution characteristics of the community at each time point should be extracted. To obtain the simple parameters from the experiments or the simulation, we performed the following steps. Two-dimensional information was abstracted into one-dimensional information from the diagonal region such that the number of bacteria in each subregion in the diagonal region could be obtained. The proportional distribution matrix was then obtained by calculating the ratio of the number of bacteria in each subregion to the entire diagonal area from the experimental data. To simplify the experimental distribution, we calculated the distances between the experimental distribution and all the distributions that were set in the simulation (see the section titled 'Calculation of the critical density of different distributions using the model'). The distribution in the simulation with the least distance to the experimental distribution is called the standard distribution for the experimental data. The standard distribution features extracted using this method were qualitatively consistent with the image features (Supplementary Fig. 14).

### Conversion of the number of bacteria and OD$_{600}$.
The density of the bacterial culture and the number of bacteria in the culture area had the following simple conversion relationship:

$$\text{Number of bacteria} = \sum \rho_i * V_i = \bar{\rho} * V$$

where $V_i$ is the volume of the selected culture area (in mL) and $\rho_i$ is the bacterial density in the selected culture area in proportion to the calculated $OD_{600}$. $\bar{\rho}$ and $V$ are the average density and the total volume, respectively. When the calculated $OD_{600}$ value is 1, the corresponding $\bar{\rho}$ value is ~$5 \times 10^8$ cells mL$^{-1}$. The total volume $V$ in the experiment (1600-μm size chamber) is ~$1600 \times 1600 \times 18$ μm$^3$ $-(126 \times 126)/2 \times 4 \times 18$ μm$^3$–$4.6 \times 10^{-5}$ mL (without taking into account the four corner areas). Thus, in the 1600-μm size chamber, we used 23,000 × (calculated $OD_{600}$) to determine the approximate number of bacteria.

### Statistics and reproducibility.
Unpaired $t$-test (two-tailed) was used to determine difference between means of varying conditions after it was determined that the variance was similar between groups. The data are presented as the mean ± standard error (SE). All the statistics are obtained by more than three examples.

### Reporting summary.
Further information on research design is available in the Nature Research Reporting Summary linked to this article.

### Data availability
Data related to this study have been deposited in the following Figshare repositories: https://figshare.com/s/7a8acaa599df05436b2a, https://figshare.com/s/e005210af1fbffb85762, https://figshare.com/s/e069c1995f09e0b30765, and https://figshare.com/s/0ca789b6a5c7380270e7. The source data underlying the graphs presented in all figures are in the Supplementary Data 1. The datasets generated during and/or

analyzed during the current study are available from the corresponding author upon reasonable request.

### Code availability
The Matlab codes in the current study are available from the corresponding author on reasonable request.

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

## Acknowledgements

The authors thank Shujing Wang for her help during the microfluidic device fabrication process and Fengyu Zhang, Chunhong Liu, Gaomin Feng, Xiaowen Xie, Sida Wang, Zhibo Zhang, Feng Liu, Yugang Wang, and Gen Yang for the constructive suggestions. This research was supported by the National Natural Science Foundation of China (11674010, 11434001), the National Key Research and Development Project (SQ2018YFA090070-03) and the National Center for Protein Sciences of China.

## Author contributions

C.X.L. designed the study; L.J.L., Y.W. and M.R. performed the experiments; L.J.L., T.W. and Y.W. developed the model; all the authors analyzed the data; L.J.L, C.X.L., Q.O.Y. and Y.K. wrote the paper.

## Additional information

**Competing interests:** The authors declare no competing interests.

