## [Peer Review File · Communications Biology]

Reviewers' comments:

Reviewer #1 (Remarks to the Author):

In the manuscript "Spatial cooperation in a population of mutually beneficial bacteria enhances the colonial antibiotic resistance" the authors design a mutualistic/cooperative mixture of two bacterial strains. One can grow on lactose and produces nutrients that the other one can utilize and the other degrades an antibiotic in the environment (ceftriaxone). They do find that in the absence of any other carbon source than lactose and presence of ceftriaxone both bacteria can only grow in a mixture together. In addition, the report on an inoculum dependent success of this interaction and a spatial structure of the bacterial community that spontaneously establishes in a two-dimensional environment when nutrients and antibiotics are diffusing from single points into a growth chamber with antibiotic degrading strains forming a barrier towards the source of the antibiotics and the nutrient releasing bacteria grow behind.

The manuscript gives interesting insights on the importance of spatial structures of microorganism communities that arise in response to environmental factors and might allow them to adjust to challenging conditions in cooperative interactions. While I am not qualified to assess whether the current state of the field was cited sufficiently and the mathematical modelling of the bacterial growth/survival and formation of spatial structures was done appropriately, I find the overall results interesting, in principle convincing and think they should stimulate the consideration of the spatial and temporal structure of microbial communities beyond the pure consideration of just absolute and relative composition.

However, I find especially the reporting on experimental replication and presentation of the data in the manuscript lacking. Thus, while I find the results convincing from a logical standpoint, I am not fully convinced by the presentation of the results. I find this especially problematic, as the authors signed in the reporting summary that sample sizes, description on statistical parameters and hypothesis testing are presented. However, for most experiments it is not clear to me, whether they were replicated and if so, how often. Multiple graphs contain average values and whiskers to denote variation, but the manuscript does not offer information which parameters are presented by these whiskers. Also no statistics were performed for the comparison of growth/survival rates in Fig 5A, Fig S9, Fig S10, Fig S13, Fig S15. Overall, several graphs need better detailed explanation, including clarification from which experiments (standard culturing containers, the microfluidic chips, simulations) the data is derived, especially as some graphs contain data from different experiments. In addition, for some experiments the authors showed only exemplary data, either one of multiple replicates (?) or only one time point out of a longer time series. I would encourage either inclusion of all the data in the experiment, or deposition in a publically accessible archive and linking it in the manuscript. Detailed comments on all these points can be found in my comments below.

The manuscript is generally written well, if sometimes a bit complicated. While the structure of the manuscript deviated from a strict separation into methods and results, I found this helpful to follow the reasoning of the single experiments and their incremental insights.

Thus, I would recommend to reject the manuscript in its current version, but offer the possibility to resubmit in case all the critical points can be addressed sufficiently.

Detailed comments:

Do not constantly change between symbiotic mutualistic and cooperative as the terms have somewhat different definitions. I would generally refrain from calling the system symbiotic as that term assumes a long term, but (depending on the definition) not necessarily a mutually beneficial association while

mutualistic does not include the long term association.

Clearly highlight in the single graphs which data is derived from experimental measurements vs. which is derived from simulations. At the moment, it is easy to lose track of that.

L23-25: What is "heterotrophic distribution" supposed to mean? Also the next sentence does not help in clarifying this.

L56-89: Please clarify more under which conditions which of the two mentioned bacteria are meant to be parasitic and symbiotic.

L65-68: Sentence is not completely clear, please rephrase.

L71: Sentence incomplete. Is it supposed to read "the function of [spatial distribution?] and the dynamic changes in the spatial distribution"?

L117: What is meant by "fence"? A membrane that allows nutrients to diffuse, but not bacteria to move into the reservoir?

L128: The use of the fluorescein needs to be explained a little bit more here.

L139: Please specify the position within the Appendix.

L139: How was this experiment to verify that the N bacteria did not evolve any resistance to ceftriaxone performed? With the starting stock that was also used to inoculate the co-culture experiments, or with the re-isolated N-bacteria from the finished co-culture experiment? Only the second experiment would demonstrate that no resistance within the N bacteria arose. The first one would assume identical parallel evolutionary transitions in two replicated experiments, which might not be unlikely, but is not given!

L159: I don't see how the experiment displayed in Figure S3 helps to demonstrate that a certain initial inoculum is necessary to establish a surviving mutualistic culture.

L165: I assume the average reader will not be aware what a high order Hill function is. I would suggest to replace or supplement it with the term "sigmoidal shape".

L173: This cutoff seems a bit unprecise, leading also to an unprecise answer; the OD range 1-2 is quite wide. It should be possible to fit a logistic function and thereby determine the cutoff more tractable and precise.

L181 / Fig S4: Please show the entire series of micrographs for all available times and antibiotic concentrations to assess when the distribution became uneven.

L194: Similar to above comments, were these experiments replicated and please present the data of these experiments in any way.

L200-214: In my opinion this part belongs to the methods.

L228: please change to: ...this population is more likely to die out.

L242-245: Please explicitly refer to the supplement where the equation for the population mathematical model can be found!

L273 I do not understand what the difference between the experiment shown in fig 5a and S9 is! Please clarify.

L275 Same as above. Please clarify what the difference between the experiments in Fig 5C and S10 is

L290 Same as above. Please clarify what the difference between the experiments in Fig 5C (right) and S11 is.

L313-324 Please avoid the word "evolve" in this paragraph when referring to changes of the trajectories of population growth as they are most likely not based on evolutionary changes within the bacterial cultures but rather changes in bacterial strain composition and distribution.

Fig 1 Replicates? Please make clear that the growth measurements in Fig 1A are not done in the microfluidic chip in Fig 1C, but in standard cultivation tubes. Does figure 1B show a single replicate? Was this experiment replicated? Or does it show averages of replicates?

Fig 2A How often was this experiment replicated? Please show the replicates (in the supplement) to be able to assess the generality of these patterns.

Fig 2B What do the whiskers indicate? Please add to figure legend. It seems that all measurements were done in duplicates. Is this correct? Please add this information to the graph.

Fig 3 Refer to Figure S5 to make clear how the micrograph was obtained, or better include FigS5 in Fig3.

Are the displayed values derived from a single replicate; was this experiment replicated? Indicate here as well that the four corners of a chip were used as technical replicates / to average the given values as explained in the supplement.

Fig 5 What do the whiskers of the bars indicate? Please include a statistical for differences between the different survival experiments.

Fig 7 Cut "Statistic of the" in the figure title, as the data is presented and not statistical values derived from tests on the data. Please add information on what the whiskers represent. Also please clarify whether the single dots represent all experiments performed with an OD between 0.5 and 1, 1 and 2, etc... Why was a different display chosen, two different graphs Fig7 and S17 with only the average values and the single measurements, and not a graph as in Figure 2 combining both?

S65-72 Were environmental conditions for the diffusion testing similar to the final use in the microfluidics chip? Please give conditions regarding temperature, shaking, humidity!

S76 Please add details for fluorescence microscopy.

S80 Please include information on how bacteria growth areas were identified or refer to Line S114ff

S333 I cannot follow this conversion between OD and bacteria number. First, the text does not refer to which culture are volume is exactly meant. In any case that means this $V = 5 \times 10^8 \text{ mL} = 5 \times 10^5 \text{ L}$. I doubt very much that any reaction container with a volume of 50.000 L was used! Please correct!

Table S6 This table needs more explanation!

Fig S1 Were these experiments replicated?

Fig S3 Please Indicate clearly which R^2 belongs to the red / N bacteria and green / R bacteria

Fig S4 Please show entire time series and replicates.

Fig S6 Does Growth rate refer to the value that is otherwise called survival rate?

Fig S7 Please repeat in the figure legend what N3 means.

Fig S9 What do the whiskers of the bars indicate? Please include a statistical for differences between the different survival experiments.

Fig S10 What do the whiskers of the bars indicate? Please include a statistical for differences between the different survival experiments.

Fig S15 Were these measurements replicated? Please include a statistical for differences between the different survival experiments.

Reviewer #2 (Remarks to the Author):

GENERAL

This manuscript by Lingjun Li et al. uses an elegant diffusion-based microfluidic system to show that 2 strains of E.coli that perform cooperative functions of feeding and protection, one serving as nutritional bacteria (N, which can utilize lactose and secretes intermediate products, RFP labeled) and the other as resistant bacteria (R, which expresses a beta-lactamase enzyme and is lactose-deficient, GFP labeled), forms a mutualistic population/functional symbiosis.

Under antibiotic pressure, this bacterial population adjusts its spatial distribution and the bacterial ratio to promote their survival.

The study is well performed and provides clues about how spatial cooperation in a mutualistic population enhances the colonial antibiotic resistance. The claims are novel and appeal to a wide audience.

I only have minor comments:

page 5 line 153

"we thus suspected" maybe replace with "we thus reasoned"

page 5 line 155

"to test this speculation" maybe replace with "to test this hypothesis"

page 5 line 228

is more likely to die out instead of "likely die out"

page 11 line 348

"did not be reached" should be replaced by has not been reached or was not reached.

page 12 line 365

Microbial community on its stability, artificially setting: please include a comma for better reading.

page 20 Fig. 3C text

please include space between "N bacteria and"

Table S2

did I misunderstood something here? Sensitive bacteria are DH5alpha (lactose-deficient) but they grow in M9+lactose?

Table S5

Please rearrange a bit the text in the columns strains and growth results: maybe just center text for growth results!

Reviewer #3 (Remarks to the Author):

Li et al. describe the spatial influence on cooperation between two mutualistic bacteria in a microfluidic device in a manuscript entitled "Spatial cooperation in a population of mutually beneficial bacteria enhances the colonial antibiotic resistance".

For that purpose they created a microfluidic chip in which they measured the growth of two *E. coli* strains, one of which is resistant to the given antibiotic (called R, for resistance), while the other one is able to use the sole provided energy and carbon source – lactose – and provides nutrients to the R strain (thus called the N strain). Several models are used to describe the spatial behavior in the restricted space of the microfluidic device and the main outcome is that the spatial variation of the two strains is dependent on several factors and might effect the ecology and evolution of mutualistic communities greater than thought before.

General comments:

The manuscript is well written and provides some interesting results. At some points I got a bit confused, which might has something to do with the complex matter. The introduction and the discussion are a bit too general and should be more specific on the two main topics presented herein, the microfluidic device used here and the nature and effect of the spatial distribution. For both points I

found it hard to decipher what is new here and what are the key differences to the earlier work cited here as well (e.g. ref 25). Several introductory and discussion parts are found in the results section. Some of the methods presented in the supplement should be given in the main text. Figures with statistic descriptions should be presented more appropriate. The order of the figures in the supplement should be in the order of appearance in the main text to allow better readability.

Specific comments:

Line 39 – 63 – This very general part could be shortened a bit, instead include more specific introduction later

Line 66 – do you mean provided?

Line 73 – Here you should include your previous work on that topic, Ref 25 and what was found and what leads you to the new steps here.

Line 86 – When the strain contains an additional plasmid, is not wild-type anymore. Rephrase.

Line 94 – I expected a reference on the two strains here, instead it is on yeast. Either rephrase or cite correct literature

Line 96 – Control experiment?

Line 97 – delete specific “in M9 medium with...” or/and delete the details, since they should be given in M&M

L 101 to 102 – this does not get clear here and I got confused. Which strains and why do you add this experiment here? Also clean up the supplement and add maybe add a bit text to describe the control experiment and don't use a new page for every small table. I also don't know what you mean with “SI Appendix” – as far as I can see the SI has no appendix.

L 104 to 109 is introduction.

L 122 – Where exactly do you show the uniform distribution? I guess in the bottom, center? However, this is difficult to see here. Also, include that in the legend.

L 142 – Do you have a growth curve instead of the SI table?

L 151 – How can you tell the bacteria are dead from the picture? I guess it is more like no growth.

L174 – Is OD 1 to 2 not a very high initial density range? Can you compare with literature in similar devices and discuss this in the discussion part? Is it only because of the high antibiotic dose?

L 184 – I suggest to move the S4 figure to the main text to better compare with 2A. What is low antibiotic concentration? Why do you cite a reference (27) here? This is something for the discussion. Or at least make a new sentence what is stated in the Ref.

L 187/8 – “The” symbiotic communities; “The” N bacteria. Why a new paragraph? This adds directly to the previous observation, not?

L 247 and elsewhere – “The” N and “the” R bacteria

L250 – N2 reads like nitrogen, better choose something else. “Aid” is not a good word here, since they don't grow at all otherwise. Which intermediates are released under the given conditions – are there any references on that?

L297 – Please rephrase the header, this is too long and difficult to read

Discussion - L 354 ff

The discussion reads more like a conclusion and contains more or less just general points. Some of the discussion is in the results part and in general the results are not too much discussed. Also, only few references are cited in a field which emerged from 2010 on and should provide more for discussion than just the eight cited references.

Figures:

Figure 2B: The average is hard to read – where are the x-axis boundaries here? Also, adding a box plot (while including more transparent dots) would give more information here. Possibly try violin plots as well.

Figure 4C – E are not cited in the text

Figure 5 – I don't get the statistics here, why the error bar only on the black bars and why bars at all?

Figure 6 – Why the total community as inset, which is hard to read?

Figure 7: See fig 2B – why don't use box or violin plots here?

References: Some informations – volume, pp (especially for the PNAS publications) are lacking

Tobias Goris

Reviewers' comments:

Reviewer #1 (Remarks to the Author):

In the manuscript “Spatial cooperation in a population of mutually beneficial bacteria enhances the colonial antibiotic resistance” the authors design a mutualistic/cooperative mixture of two bacterial strains. One can grow on lactose and produces nutrients that the other one can utilize and the other degrades an antibiotic in the environment (ceftriaxone). They do find that in the absence of any other carbon source than lactose and presence of ceftriaxone both bacteria can only grow in a mixture together. In addition, the report on an inoculum dependent success of this interaction and a spatial structure of the bacterial community that spontaneously establishes in a two-dimensional environment when nutrients and antibiotics are diffusing from single points into a growth chamber with antibiotic degrading strains forming a barrier towards the source of the antibiotics and the nutrient releasing bacteria grow behind.

The manuscript gives interesting insights on the importance of spatial structures of microorganism communities that arise in response to environmental factors and might allow them to adjust to challenging conditions in cooperative interactions. While I am not qualified to assess whether the current state of the field was cited sufficiently and the mathematical modelling of the bacterial growth/survival and formation of spatial structures was done appropriately, I find the overall results interesting, in principle convincing and think they should stimulate the consideration of the spatial and temporal structure of microbial communities beyond the pure consideration of just absolute and relative composition.

However, I find especially the reporting on experimental replication and presentation of the data in the manuscript lacking. Thus, while I find the results convincing from a logical standpoint, I am not fully convinced by the presentation of the results. I find this especially problematic, as the authors signed in the reporting summary that sample sizes, description on statistical parameters and hypothesis testing are presented. However, for most experiments it is not clear to me, whether they were replicated and if so, how often. Multiple graphs contain average values and whiskers to denote variation, but the manuscript does not offer information which parameters are presented by these whiskers. Also no statistics were performed for the comparison of growth/survival rates in Fig 5A, Fig S9, Fig S10, Fig S13, Fig S15. Overall, several graphs need better detailed explanation, including clarification from which experiments (standard culturing containers, the microfluidic

chips, simulations) the data is derived, especially as some graphs contain data from different experiments.

In addition, for some experiments the authors showed only exemplary data, either one of multiple replicates (?) or only one time point out of a longer time series. I would encourage either inclusion of all the data in the experiment, or deposition in a publically accessible archive and linking it in the manuscript. Detailed comments on all these points can be found in my comments below.

Our answer: Thank you for the referee's comments. We modified the paper according to the reviewer's suggestions. We have included the most important data in the revised paper, with experiments having been repeated at least three times. In addition, we have added additional statements and statistical information in the text and more details in the SI.

The manuscript is generally written well, if sometimes a bit complicated. While the structure of the manuscript deviated from a strict separation into methods and results, I found this helpful to follow the reasoning of the single experiments and their incremental insights.

Thus, I would recommend to reject the manuscript in its current version, but offer the possibility to resubmit in case all the critical points can be addressed sufficiently.

Our answer: We would like to thank the reviewers for their many thoughtful questions and constructive comments/suggestions. We have taken into account the reviewers' comments and made modifications to our original manuscript. In the following section, we provide a detailed account of the most important modifications, and we would like to resubmit the revised manuscript for your consideration.

Detailed comments:

Do not constantly change between symbiotic mutualistic and cooperative as the terms have somewhat different definitions. I would generally refrain from calling the system symbiotic as that term assumes a long term, but (depending on the definition) not necessarily a mutually beneficial association while mutualistic does not include the long term association.

Our answer: Thank you for the comments. We replaced the term "symbiotic" with "mutualistic" in the revised manuscript.

Clearly highlight in the single graphs which data is derived from experimental measurements vs. which is derived from simulations. At the moment, it is easy to lose track of that.

Our answer: Thank you for the comments. Based on the suggestion, we discuss the data sources in the legends of Fig. 2B, Fig. 3, Fig. 5C and Fig. 6 in the revised manuscript.

L23-25: What is “heterotrophic distribution” supposed to mean? Also the next sentence does not help in clarifying this.

Our answer: Thank you for the comments. The word used in this article is “heterotopic distribution”, not “heterotrophic distribution”. Our intention is to show that the spatial distributions of the two bacterial strains are not the same, with each one having its own characteristics. In the revised manuscript, we modified “a heterotrophic distribution” to “the distributions” to better communicate this point.

L56-89: Please clarify more under which conditions which of the two mentioned bacteria are meant to be parasitic and symbiotic.

Our answer: Thank you for the comments. We have modified the description of this section in the revised manuscript (page 2).

“For example, bacteria that can provide nutrients and those that can degrade antibiotics may be mutualistic in a nutrient-deficient environment in the presence of antibiotics. However, when the antibiotic pressure is removed, the growth of the former bacteria does not depend on the latter, at which time there may be a parasitic relationship between the two bacteria.”

L65-68: Sentence is not completely clear, please rephrase.

Our answer: Thank you for the comments. We have modified the description of this section in the revised manuscript (page 2) as follows:

“Kim *et al.* generated a mutualistic community of three bacterial strains, each of which was capable of either degrading antibiotics, providing a carbon source or providing a nitrogen source. A community could not develop when the three strains were uniformly mixed and cultured together, with maximal growth only achieved when they were separated by a specific distance.”

L71: Sentence incomplete. Is it supposed to read “the function of [spatial distribution?] and the dynamic changes in the spatial distribution”?

Our answer: Thank you for the comments. The referee's understanding is correct, and we have supplemented this sentence in the revised manuscript (page 2-3) based on the comments as follows: "However, due to the limitations imposed by experimental conditions, previous studies on the characteristics of the mutualism were mainly restricted to analyses of the bacterial species, quantity, and proportion in a community and did not explore the function of the spatial distribution and the dynamic changes in the spatial distribution."

L117: What is meant by "fence"? A membrane that allows nutrients to diffuse, but not bacteria do move into the reservoir?

Our answer: Thank you for the comments. The referee's understanding of "fence" is correct. In the revised manuscript, we have described the fence-like channels in both the chip schematic and in the microscopic images of the chip in Figure 1C. After filling the fence-like channels with gel, the solidified gel in the fence-like channels acts as a semi-permeable membrane. We modified the text by changing the term "fence" to "fence-like channels" and "gel filled channels" in Fig. 1C.

L128: The use of the fluorescein needs to be explained a little bit more here.

Our answer: Thank you for the comments. We have added a brief description of the fluorescein diffusion experiment in the text (page 4). More details regarding the experiment can be found in the SI – Section 5.

"To observe the diffusion characteristics of the substance in the chip, in addition to the simulation calculation, we also used fluorescein for the diffusion experiment, which has a diffusion coefficient that is close to that of lactose and CRO (SI – Section 5)."

L139: Please specify the position within the Appendix.

Our answer: Thank you for the comments. We have checked the text and added specific information in the revised manuscript (page 5) as follows:

"We also ruled out the possibility that bacterial strains that do not establish a functional mutually beneficial relationship would develop resistance in the chip (SI – Section 7 and Table S4)."

L139: How was this experiment to verify that the N bacteria did not evolve any resistance to

ceftriaxone performed? With the starting stock that was also used to inoculate the co-culture experiments, or with the re-isolated N-bacteria from the finished co-culture experiment? Only the second experiment would demonstrate that no resistance within the N bacteria arose. The first one would assume identical parallel evolutionary transitions in two replicated experiments, which might not be unlikely, but is not given!

Our answer: Thank you for the comments. First, we used the N bacteria from the starting stock in each experiment. We observed that the N bacteria can evolve resistance to ceftriaxone if the mutualistic communities were cultured for a long time. Specifically, when the two bacteria were co-cultured in the chip for over 100 hours, the N bacteria sometimes grew rapidly in the area close to the antibiotic source instead of the center. This phenomenon is likely associated with the evolution of drug resistance. However, within 50 hours, this phenomenon did not occur, and all experiments exhibited similar behaviors with the N bacteria grew in the center area of the chamber. Therefore, we believe that in the short-term culture, which is used in our experiments (approximately 50 h), the development of the N bacteria occurs at an extremely low frequency and can be ignored.

L159: I don't see how the experiment displayed in Figure S3 helps to demonstrate that a certain initial inoculum is necessary to establish a surviving mutualistic culture.

Our answer: Thank you for the comments. Fig. S3 is a supplement to the “quantitatively analyzed the experimental results”. Fig. S3 shows the linear relationship between fluorescence intensity and the calculated OD₆₀₀, showing that the fluorescence intensity can be used to determine the density of bacteria in the chip. We modified the text (page 5) as follows:

“...quantitatively analyzed bacterial population density by assessing the fluorescence intensity (*SI* – Section 9, Fig. S3).”

L165: I assume the average reader will not be aware what a high order Hill function is. I would suggest to replace or supplement it with the term “sigmoidal shape”.

Our answer: Thank you for the comments. In the revised manuscript, we have revised the text (page 6) to read as follows:

“...and their qualitative relationships were similar to a high-order Hill function (or sigmoidal function)...”

L173: This cutoff seems a bit unprecise, leading also to an unprecise answer; the OD range 1-2 is quite wide. It should be possible to fit a logistic function and thereby determine the cutoff more tractable and precise.

Our answer: Thank you for the comments. We were uncertain as to what type of function would be suitable in this instance. However, the data obtained from the experiment is not accurate enough due to the experimental variations and limited repetitions. We did fit the data with the Hill function $Y=a+b \times x^n / (x^n+x_0^n)$, which gives a x_0 value of approximately 1.5, which is comparable to the simulation result. However, we decided that using a range to indicate the transition range between the growth and death states is more suitable in our experiment. This point has little influence on our future discussion regarding the distribution of ratio effects. Moreover, when comparing the resistance of mutualistic communities to those cultured in different sized spaces, such a range is sufficient for comparison.

L181 / Fig S4: Please show the entire series of micrographs for all available times and antibiotic concentrations to assess when the distribution became uneven.

Our answer: Thank you for the comments. In the new Fig. 2A, Fig. S4 and Fig. S5, we have added an additional time series of micrographs and more samples with different antibiotic concentrations.

Fig. S4 (A) Time series of micrographs of one sample (Fig. 3 shows one diagonal line of the sample). (B, C, D) The other three diagonal lines of the same sample as shown in Fig. S4 (A). (E) The time series of micrographs of two additional samples.

L194: Similar to above comments, were these experiments replicated and please present the data of these experiments in any way.

Our answer: Thank you for the comments. The repeatability of the mutualistic community growth experiment has been explained in the above question. Repeated experiments were also performed on the culture of parasitic communities, which are shown in the new Fig. S5.

Fig. S5 Time series of three repeated experiments with 0 $\mu\text{g/mL}$ CRO (parasitic communities).

L200-214: In my opinion this part belongs to the methods.

Our answer: Thank you for the comments. We have accepted the referee's suggestion and moved this passage to the Materials and Methods section.

L228: please change to: ...this population is more likely to die out.

Our answer: Thank you for the comments. We have accepted the suggestion and made corrections in the revised manuscript.

L242-245: Please explicitly refer to the supplement where the equation for the population mathematical model can be found!

Our answer: Thank you for the comments. We apologize for the confusion and have added the specific information to the revised manuscript (page 8) as follows:

“To quantitatively study their effect, we employed a population mathematical model based on the reaction-diffusion equation to analyze the changes in the distribution and N/R ratio of the system and to simulate the critical densities and survival rates under different conditions (SI – Section 12-13, Table S6).”

L273 I do not understand what the difference between the experiment shown in fig 5a and S9 is! Please clarify.

Our answer: Thank you for the comments and we apologize for the confusion. Fig. 5A shows the survival rate of the N bacteria, whereas Fig. S9 (Fig. S8 in the revised SI) shows the overall survival rate of the community. In the original manuscript, we only explained this difference in the legend and not within the figure. In the revised manuscript, we used n_N instead of N to indicate the number of the N bacteria in the figure and used n_{All} to indicate the total number of bacteria in the community.

L275 Same as above. Please clarify what the difference between the experiments in Fig 5C and S10 is

Our answer: Thank you for the comments. The changes we have made are the same as those described in the response to the above question.

L290 Same as above. Please clarify what the difference between the experiments in Fig 5C (right) and S11 is.

Our answer: Thank you for the comments. The changes we have made are the same as those described in the response to the above question.

L313-324 Please avoid the word “evolve” in this paragraph when referring to changes of the trajectories of population growth as they are most likely not based on evolutionary changes within the bacterial cultures but rather changes in bacterial strain composition and distribution.

Our answer: Thank you for the comments. In the revised manuscript, we replaced “evolve” in the text with “change”.

Fig 1 Replicates? Please make clear that the growth measurements in Fig 1A are not done in the

microfluidic chip in Fig 1C, but in standard cultivation tubes. Does figure 1B show a single replicate? Was this experiment replicated? Or does it show averages of replicates?

Our answer: Thank you for the comments. The experiment in Fig. 1B was performed in standard cultivation tubes. The original image in Fig. 1B shows a single replicate. We have replaced these results with those showing averages of seven to eight repeated experiments by using a microplate reader. The results show a similar behavior as that previously observed.

We have added a description of the experimental conditions in the legend (page 17) as follows:

“The dots showed the averages of seven to eight repeated experiments and the whiskers indicated the standard deviation. The experiment was performed in 96-well plates.”

Fig 2A How often was this experiment replicated? Please show the replicates (in the supplement) to be able to assess the generality of these patterns.

Our answer: Thank you for the comments. The results shown in Fig. 2A were ubiquitous. As mentioned in the answer above, we have added Fig. S4 and Fig. S5 to show more repetitive results.

Fig 2B What do the whiskers indicate? Please add to figure legend. It seems that all measurements were done in duplicates. Is this correct? Please add this information to the graph.

Our answer: Thank you for the comments. For each sample point, only one set of measurements was shown, which was the data represented by the black dots in the graph. Based on the suggestion of another referee, we changed the dot plot of the average value to a box plot that shows more information. In the figure, the red line indicates the maximum and minimum values in the 1.5 IQR range.

We have modified the legend in the revised manuscript (page 18) as follows:

“The superimposed box plot is drawn in red.”

Fig 3 Refer to Figure S5 to make clear how the micrograph was obtained, or better include FigS5 in Fig3.

Our answer: Thank you for the comments. According to the suggestion, we have integrated the original Fig. S5 into Fig. 3 in the revised manuscript (page 19).

Are the displayed values derived from a single replicate; was this experiment replicated? Indicate here as well that the four corners of a chip were used as technical replicates / to average the given values as explained in the supplement.

Our answer: Thank you for the comments. Fig. 3 showed the results of a single sample, but there were many similar repeatable results. More sample information has been added to Fig. S4, with Fig. S4 (E) showing photomicrographs of two additional samples.

In addition, the growth characteristics of the four diagonal lines of the same sample were similar. We added a micrograph of the entire culture area in Fig. S4 (A) and a diagonal screenshot of the other three corners in Fig. S4 (B), (C), and (D).

Fig 5 What do the whiskers of the bars indicate? Please include a statistical for differences between the different survival experiments.

Our answer: Thank you for the comments. The whiskers indicate the standard deviation of multiple samples with the same characteristics. We have added information describing the significance of the differences between samples with different characteristics and have added the above information to the legend (page 21) as follows:

“The whisker represents the standard deviation of all samples with one characteristic. The asterisk indicates the significance of the difference between the samples in each group. “*” indicates that the P value of the t test is less than 0.05, and “**” indicates that the P value of the t test is less than 0.01. There are 3 to 5 replicate samples in each group.”

Fig 7 Cut “Statistic of the” in the figure title, as the data is presented and not statistical values derived from tests on the data. Please add information on what the whiskers represent. Also please clarify whether the single dots represent all experiments performed with an OD between 0.5 and 1, 1 and 2, etc... Why was a different display chosen, two different graphs Fig7 and S17 with only the average values and the single measurements, and not a graph as in Figure 2 combining both?

Our answer: Thank you for the comments. In the original manuscript, the data for each sample and the average value were displayed separately because we believed that it would be difficult to visualize if all of the data were displayed together. We combined Fig. 7 and Fig. S17 according to

the referee's suggestion and revised the title and legend of Fig. 7. In addition, based on the suggestion of another referee, we changed the dot plot of the average value to a box plot that shows more information. We also added additional information to the legend (page 23) as follows:

“The x-axis boundaries of the superimposed box occur at OD₆₀₀ values of 0.5-1, 1-2, 2-4, 4-8, 8-16 and 16-32. The x-coordinate of each box is the average of the x-coordinate of the samples it contains.”

S65-72 Were environmental conditions for the diffusion testing similar to the final use in the microfluidics chip? Please give conditions regarding temperature, shaking, humidity!

Our answer: Thank you for the comments. The conditions of the diffusion experiment were the same as those used for the final use of the chip. The temperature was 37 °C, and cotton balls were used for moisturization. Because the spatial distribution of the community could not be affected by external forces, no shaking was performed in all microfluidic experiments. We have added the above information to the revised SI (page 3) as follows:

“The environmental conditions used for the diffusion testing (37 °C, no shaking and the use of moisturized cotton balls) were similar to the final conditions used in the microfluidic chip.”

S76 Please add details for fluorescence microscopy.

Our answer: Thank you for the comments. We have added the manufacturer and model information for the fluorescence microscope to the revised SI (page 3) as follows:

“We added 1 mM FITC to the reservoir and then used a fluorescence microscope (Nikon Eclipse Ti-E) to obtain images every 20 min.”

S80 Please include information on how bacteria growth areas were identified or refer to Line S114ff

Our answer: Thank you for the comments. We have added additional details in the revised SI (page 3 and page 4) as follows:

“For each image, the boundaries of the culture area were determined using the bright channels, and the culture area was then fixed in the same position for subsequent imaging.”

“First, we selected the culture area from the image for analysis. Because the position of the culture area is fixed for each image, we were able to use MATLAB to evaluate the information in the same area.”

S333 I cannot follow this conversion between OD and bacteria number. First, the text does not refer to which culture are volume is exactly meant. In any case that means this V= has to be $5 \times 10^8 \text{ mL} = 5 \times 10^5 \text{ L}$. I doubt very much that any reaction container with a volume of 50.000 L was used! Please correct!

Our answer: Thank you for the comments. We apologize for failing to explain the conversion between the number of bacteria and the OD₆₀₀ value. We have made the following corrections to better show this conversion (SI page 9):

$$\text{“Number of bacteria} = \sum \rho_i * V_i = \bar{\rho} * V,$$

where V_i is the volume of the selected culture area (in mL) and ρ_i is the bacterial density in the selected culture area in proportion to the calculated OD₆₀₀. $\bar{\rho}$ and V are the average density and the total volume, respectively. When the calculated OD₆₀₀ value is 1, the corresponding $\bar{\rho}$ value is approximately 5×10^8 cells/mL. The total volume V in the experiment (1600- μm size chamber) is approximately $1600 \times 1600 \times 18 \mu\text{m}^3 - (126 \times 126) / 2 \times 4 \times 18 \mu\text{m}^3 \sim 4.6 \times 10^{-5} \text{ mL}$ (without taking into account the four corner areas). Thus, in the 1600- μm size chamber, we used $23,000 \times (\text{calculated OD}_{600})$ to determine the approximate number of bacteria.”

Table S6 This table needs more explanation!

Our answer: Thank you for the comments. We have added additional information to the revised SI (page 13), and we have separated the merged photographs to individual GFP and mCherry channels for the co-culture condition.

“Note: The images are superimposed with the GFP and mCherry channels, and the LUTs of the images taken at the same time are consistent. The resistant bacteria are labeled with GFP, and the nutritional bacteria are labeled with RFP. The medium used was M9 containing 0.8% lactose and 250 $\mu\text{g/mL}$ Amp. Images of co-cultured sample are represented by each single fluorescent channel and a combination of two channels.”

Fig S1 Were these experiments replicated?

Our answer: Thank you for the comments. The image in the original SI did not show duplicate results, and we replaced it with a new one showing repeated experiments.

Fig S3 Please Indicate clearly which R² belongs to the red / N bacteria and green / R bacteria

Our answer: Thank you for the comments. Based on the suggestions, we have made the appropriate changes in the revised SI.

Fig S4 Please show entire time series and replicates.

Our answer: Thank you for the comments. We have added some micrographs of additional time points in Fig. 2A based on the advice of another referee. In addition, as mentioned in the answer above, the entire time series and replicates are shown in the new Fig. S4 and Fig. S5.

Furthermore, we also added a sentence to the revised manuscript (page 6) as follows:

“Additional images of entire time series and samples of the two conditions are provided in the Supporting Information (Figs. S4 and S5).”

Fig S6 Does Growth rate refer to the value that is otherwise called survival rate?

Our answer: Thank you for the comments. Growth and survival rates are different and are described in detail in the Methods section. The survival rate at a certain time is the ratio of the number of bacteria at that time to the number of bacteria at 0 h. In the calculation of the growth rate, time information is added and the ln function is used. Both can separate the grow state of the bacteria from the initial state, but for the comparison between different time points, the growth rate that considers the time periods is better than the survival rate.

Fig S7 Please repeat in the figure legend what N3 means.

Our answer: Thank you for the comments. The meaning of N3 has been added to the legend of Fig. S6 (SI page 20).

“N3 is the nutrient released by dead bacteria that can be used by living bacteria.”

Fig S9 What do the whiskers of the bars indicate? Please include a statistical for differences between the different survival experiments.

Our answer: Thank you for the comments. The whiskers indicate the standard deviation of multiple samples with the same characteristics. We have added this information to the legend in the revised

manuscript. A description of the significance of the differences between samples with different characteristics was also added to Fig. S9 (Fig. S8 in the revised SI).

Fig S10 What do the whiskers of the bars indicate? Please include a statistical for differences between the different survival experiments.

Our answer: Thank you for the comments. The whiskers indicate the standard deviation of multiple samples with the same characteristics. We have added this information to the legend (Fig. S9 in the revised SI). Because the P values between the results of each group were greater than 0.05, no asterisk was included in the figure.

Fig S15 Were these measurements replicated? Please include a statistical for differences between the different survival experiments.

Our answer: Thank you for the comments. These results were all calculated by simulation. When the input parameters were unchanged, the calculation results were the same each time.

Reviewer #2 (Remarks to the Author):

GENERAL

This manuscript by Lingjun Li et al. uses an elegant diffusion-based microfluidic system to show that 2 strains of E.coli that perform cooperative functions of feeding and protection, one serving as nutritional bacteria (N, which can utilize lactose and secretes intermediate products, RFP labeled) and the other as resistant bacteria (R, which expresses a beta-lactamase enzyme and is lactose-deficient, GFP labeled), forms a mutualistic population/functional symbiosis.

Under antibiotic pressure, this bacterial population adjusts its spatial distribution and the bacterial ratio to promote their survival.

The study is well performed and provides clues about how spatial cooperation in a mutualistic population enhances the colonial antibiotic resistance. The claims are novel and appeal to a wide audience.

I only have minor comments:

page 5 line 153

“we thus suspected” maybe replace with “we thus reasoned”

Our answer: Thank you for the comments. We have replaced this passage in the revised manuscript based on this comment.

page 5 line 155

“to test this speculation” maybe replace with “to test this hypothesis”

Our answer: Thank you for the comments. We have replaced this passage in the revised manuscript based on this comment.

page 5 line 228

is more likely to die out instead of “likely die out”

Our answer: Thank you for the comments. We have replaced this passage in the revised manuscript based on this comment.

page 11 line 348

“did not be reached” should be replaced by has not been reached or was not reached.

Our answer: Thank you for the comments. We have replaced this passage in the revised manuscript based on this comment.

page 12 line 365

Microbial community on its stability, artificially setting: please include a comma for better reading.

Our answer: Thank you for the comments. We have made this correction in the revised manuscript based on this comment.

page 20 Fig. 3C text

please include space between “N bacteria and”

Our answer: Thank you for the comments. We have made this correction in the revised manuscript based on this comment.

Table S2

did I misunderstood something here? Sensitive bacteria are DH5alpha (lactose-deficient) but they grow in M9+lactose?

Our answer: We apologize for the mistake. The growth results of nutritional and sensitive bacteria

were marked incorrectly. We have made this correction in the revised SI.

Table S5

Please rearrange a bit the text in the columns strains and growth results: maybe just center text for growth results!

Our answer: Thank you for the comments. We have modified the format of Tables S1-S6 in the revised SI.

Reviewer #3 (Remarks to the Author):

Li et al. describe the spatial influence on cooperation between two mutualistic bacteria in a microfluidic device in a manuscript entitled “Spatial cooperation in a population of mutually beneficial bacteria enhances the colonial antibiotic resistance”.

For that purpose they created a microfluidic chip in which they measured the growth of two *E. coli* strains, one of which is resistant to the given antibiotic (called R, for resistance), while the other one is able to use the sole provided energy and carbon source – lactose - and provides nutrients to the R strain (thus called the N strain). Several models are used to describe the spatial behavior in the restricted space of the microfluidic device and the main outcome is that the spatial variation of the two strains is dependent on several factors and might effect the ecology and evolution of mutualistic communities greater than thought before.

General comments:

The manuscript is well written and provides some interesting results. At some points I got a bit confused, which might has something to do with the complex matter. The introduction and the discussion are a bit too general and should be more specific on the two main topics presented herein, the microfluidic device used here and the nature and effect of the spatial distribution. For both points I found it hard to decipher what is new here and what are the key differences to the earlier work cited here as well (e.g. ref 25). Several introductory and discussion parts are found in the results section. Some of the methods presented in the supplement should be given in the main text. Figures

with statistic descriptions should be presented more appropriate. The order of the figures in the supplement should be in the order of appearance in the main text to allow better readability.

Our answer: Thank you for the comments. We have modified the Introduction and Discussion sections, removing some general discussions and adding more specific information. For the microfluidic chip used in ref 25, the gradient is stable when observing the bacterial behaviors, since there are source and sink wells in the device. A short observation period was used to determine the MIC value of the bacteria while the concentration gradient of the substance formed by the chip used in ref 25 was stable and not changed by bacteria. In our studies, a mutualistic community composed of two bacteria was used to observe the characteristics of the community. In addition, our device only had source wells. The concentration gradient and the activity of the bacteria would change with time, since a much higher bacterial density was used in this study compared with that used in Ref 25. Our system is more similar to natural conditions in that the bacterial mutualistic community aggregates and communicates with the outside world through limited contact surfaces (source) and the bacteria create a chemical gradient within the community by themselves.

Based on the referee's suggestion, we have moved some of the information from the Results section to the Introduction and Discussion sections. Some of the methods described in the supplementary material have been added to the text. In addition, the dot plot showing the average value was changed to a box plot with statistical description. Finally, the order of the numbers in the supplementary material was arranged in the order of appearance in the main text.

Specific comments:

Line 39 – 63 – This very general part could be shortened a bit, instead include more specific introduction later

Our answer: Thank you for the comments. We have shortened this section and added a more specific introduction (page 2-3).

Line 66 – do you mean provided?

Our answer: Thank you for the comments. We apologize for the confusion, and we have modified this section (page 2) as follows:

“Kim *et al.* generated a mutualistic community of three bacterial strains, each of which was capable of either degrading antibiotics, providing a carbon source or providing a nitrogen source. A community could not develop when the three strains were uniformly mixed and cultured together, with maximal growth only achieved when they were separated by a specific distance.”

Line 73 – Here you should include your previous work on that topic, Ref 25 and what was found and what leads you to the new steps here.

Our answer: Thank you for the comments. Based on the recommendations, we have added the citation and description for Ref 25 as follows (page 3):

“The use of microfluidic chips can provide a well-defined growth environment for microbial community studies [23,25]. For example, using hydrogel in a chip can create a zero-flow environment in which substances can be exchanged [25]. These characteristics suggested that a space enclosed by gel barriers in a chip could be designed to simulate the natural environment in which bacterial communities grow.”

Line 86 – When the strain contains an additional plasmid, is not wild-type anymore. Rephrase.

Our answer: Thank you for the comments. Based on the suggestion, we deleted the “wild type”.

Line 94 – I expected a reference on the two strains here, instead it is on yeast. Either rephrase or cite correct literature

Our answer: Thank you for the comments. The reference was intended to illustrate that, in theory, there will be a mutualistic relationship between cells with the desired attributes. At present, there no research has been performed on the community we have built. Based on the referee’s recommendations, we cited a study (Ref 16) on the formation of mutualistic *E. coli* communities through the degradation of antibiotics and an investigation (Ref 18) on the formation of mutualistic *E. coli* communities through the release of intermediate nutrients. These references were added to the Introduction section in the following passages (page 2):

“Compared with simple cooperation, such as providing nutrition or degrading antibiotics, multitype cooperative behavior is more complex and closer to the real situation under natural conditions [16–18].”

“...allowing the two strains to establish a mutualism (Fig. 1A).”

Line 96 – Control experiment?

Our answer: Thank you for the comments. We have replaced this text in the revised manuscript based on this comment.

Line 97 – delete specific “in M9 medium with...” or/and delete the details, since they should be given in M&M

Our answer: Thank you for the comments. We have changed this passage in the revised manuscript based on this comment. The description of the medium has been moved to the Materials and Methods section (page 12).

“The specific medium that allowed for mutualistic behavior between the two strains is M9 medium supplemented with lactose as the only carbon source and CRO.”

L 101 to 102 – this does not get clear here and I got confused. Which strains and why do you add this experiment here? Also clean up the supplement and add maybe add a bit text to describe the control experiment and don't use a new page for every small table. I also don't know what you mean with “SI Appendix” – as far as I can see the SI has no appendix.

Our answer: Thank you for the comments. All of the control experiments were performed to show that the bacteria containing the plasmid had the desired activity but others did not. We have changed the “*SI Appendix*” to “*SI*” and referred to the corresponding text as much as possible in the revised manuscript. We hope that these modifications help to better communicate the content of the article (page 4):

“To verify the mutualistic relationship between the two strains, we conducted a control experiment in which the two strains were cultured alone or cocultured in a specific medium in standard tubes [13]. The results showed that the cultures with only one strain did not grow, whereas the cocultures exhibited growth (Fig. 1B), indicating that these two strains form a mutualism under certain experimental conditions [16]. In addition, by designing another control experiment, we eliminated the possibility that the two strains are able to achieve mutualism by means other than the designed functions (*SI* – Section 1-3, Table S1-3, Fig. S1).”

L 104 to 109 is introduction.

Our answer: Thank you for the comments. According to the suggestion, we have moved this section to the introduction.

L 122 – Where exactly do you show the uniform distribution? I guess in the bottom, center? However, this is difficult to see here. Also, include that in the legend.

Our answer: Thank you for the comments. The referee's understanding that the uniform distribution is shown in the center of the bottom of Fig. 1C is correct. We have added the following note in the legend (page 17):

“The image in the middle of the bottom panel is a photomicrograph that was taken after the bacteria were added but before the culture was started. At this time, the distribution of the bacteria is uniform.”

L 142 – Do you have a growth curve instead of the SI table?

Our answer: Thank you for the comments. Because the method used to quantitatively determine the bacterial density in the chip was not introduced before this section, showing the growth curve here is not preferable. Furthermore, compared to the growth process, we were more concerned about the growth results after a period of time. We added the photomicrographs of the community in the chip cultured for 0 and 20 h in Table S5. By comparing the micrographs at these two time points, the growth of the community can be easily judged.

L 151 – How can you tell the bacteria are dead from the picture? I guess it is more like no growth.

Our answer: Thank you for the comments. Although the death of bacteria in the corners could be observed, it was not obvious compared to growth; thus, the referee's statement may be more accurate. Therefore, we have made the appropriate changes in the revised manuscript (page 5) as follows:

“In our experiments, however, we observed that some communities did not grow (Fig. 2A).”

L174 – Is OD 1 to 2 not a very high initial density range? Can you compare with literature in similar devices and discuss this in the discussion part? Is it only because of the high antibiotic dose?

Our answer: Thank you for the comments. This initial density range used in this study is relatively high compared with those that are typically used in tube experiments or MIC studies in similar microfluidic systems [1] but is low compared with that used in bacterial biofilms studies [2,3]. In our previous studies investigating the MIC values of bacteria in chips, it was necessary to minimize the influence of bacteria on the antibiotic concentration, and only OD₆₀₀ values of 0.1-0.2 were used for the initial density, with only a 3-4 hour growth period. However, for bacterial mutualism experiments in this study, we observed that an initial OD₆₀₀ 1-2 is needed for growth in a 1600- μ m culture system and a much high initial bacterial density (such as an OD₆₀₀ 8-16 for the 800- μ m culture system) is needed for smaller culture systems to resist to the antibiotics present in the medium. No similar studies have been performed using a similar device. Many studies have shown that bacterial biofilms in nature have a much higher bacterial density, such as 10¹⁰ cells/ml (calculated OD₆₀₀ is approximately tens to hundreds). We have add some sentences on this topic in the Discussion section (page 11) as follows:

“We observed that an initial OD₆₀₀ 1-2 was needed for growth in the 1600- μ m culture system, whereas a much higher initial bacterial density (such as an OD₆₀₀ 8-16 for the 800- μ m culture system) was needed for a smaller culture system to resist the antibiotics present in the medium. These densities are much higher compared with those observed in MIC studies [25] using a similar microfluidic system and is smaller than that of formed biofilms in other microfluidic systems (approximately 3 \times 10⁹~10¹⁰ cells/mL) [15,30].”

[1] Ran, Min; Wang, Ying; Wang, Sida; Luo, Chunxiong (2015): Pump-free gradient-based micro-device enables quantitative and high-throughput bacterial growth inhibition analysis. In *Biomed Microdevices* 17 (4), p. 1265. DOI: 10.1007/s10544-015-9971-8.

[2] Nadell, C. D.; Ricaurte, D.; Yan, J.; Drescher, K.; Bassler, B. L. (2017): Flow environment and matrix structure interact to determine spatial competition in *Pseudomonas aeruginosa* biofilms. In *Elife* 6, e21855. DOI: 10.7554/eLife.21855.

[3] Nguyen, D.; Joshi-Datar, A.; Lepine, F.; Bauerle, E.; Olakanmi, O.; Beer, K. et al. (2011): Active Starvation Responses Mediate Antibiotic Tolerance in Biofilms and Nutrient-Limited Bacteria. In *Science* 334 (6058), pp. 982–986. DOI: 10.1126/science.1211037.

L 184 – I suggest to move the S4 figure to the main text to better compare with 2A. What is low antibiotic concentration? Why do you cite a reference (27) here? This is something for the discussion. Or at least make a new sentence what is stated in the Ref.

Our answer: Thank you for the comments. First, we merged Fig. S4 into Fig. 2A. A low antibiotic concentration indicates that the CRO concentration is 0. Because the previous phrasing was not accurate enough, we replace it with “in the environment without CRO”. The model in the cited reference 27 (reference 28 in the revised manuscript) predicted the spatial distribution of bacteria with different interactions. We have added a sentence based on this reference as follows (page 6): “Additional images of the entire time series and samples of the two conditions are provided in the Supporting Information (Figs. S4 and S5). Different interactions between populations results in different spatial distribution patterns of the communities [28].”

L 187/8 – “The” symbiotic communities; “The” N bacteria. Why a new paragraph? This adds directly to the previous observation, not?

Our answer: Thank you for the comments. We have made corrections according to the suggestions. It is desirable to use segmentation to distinguish the distribution features we observed (see the above paragraph) and for the specific analysis of the distribution (see the latter paragraph). In the revised manuscript, we have merged these passages into one paragraph.

L 247 and elsewhere – “The” N and “the” R bacteria

Our answer: Thank you for the comments. We have made the appropriate corrections according to the suggestions.

L250 – N₂ reads like nitrogen, better choose something else. “Aid” is not a good word here, since they don’t grow at all otherwise. Which intermediates are released under the given conditions – are there any references on that?

Our answer: Thank you for the comments. First, we used “N₂” to replace “N₂”. Considering that “₂” was not used as a subscript, it can still be distinguished from N₂. In addition, we changed “aid the growth of the R bacteria” to “used by the R bacteria”. As explained in page 3 of the text, the

intermediates released under the specific conditions include some organic acids such as succinic acid. We have added the corresponding reference in page 3 as follows:

“To establish a feeding-protection mutualism, we selected two strains of *E. coli*: one strain contains the lac genes, which can utilize lactose and secrete intermediate products (such as succinic acid), and the other strain expresses a β -lactamase enzyme, which inactivates the antibiotic ceftriaxone sodium (CRO) but cannot utilize lactose [17].”

L297 – Please rephrase the header, this is too long and difficult to read

Our answer: Thank you for the comments. We have shortened the title to “Changes in the bacterial distribution and N/R ratio enhance the community survival rate” in the revised manuscript (page 9).

Discussion – L 354 ff

The discussion reads more like a conclusion and contains more or less just general points. Some of the discussion is in the results part and in general the results are not too much discussed. Also, only few references are cited in a field which emerged from 2010 on and should provide more for discussion than just the eight cited references.

Our answer: Thank you for the comments. We have modified the Discussion section (page 11) and added additional references as follows:

“We observed that an initial OD₆₀₀ 1-2 was needed for growth in the 1600- μ m culture system, whereas a much higher initial bacterial density (such as an OD₆₀₀ 8-16 for the 800- μ m culture system) was needed for a smaller culture system to resist the antibiotics present in the medium. These densities are much higher compared with those observed in MIC studies [25] using a similar microfluidic system and is smaller than that of formed biofilms in other microfluidic systems (approximately $3 \times 10^9 \sim 10^{10}$ cells/mL) [15,30].”

“The spatial structures observed in our experiments are different from those obtained in previous models or experiments [28,31]. This difference may be caused by the closure of the community growth space and the complexity of interactions between the populations.”

Figures:

Figure 2B: The average is hard to read – where are the x-axis boundaries here? Also, adding a box

plot (while including more transparent dots) would give more information here. Possibly try violin plots as well.

Our answer: Thank you for the comments. We have changed the dot plot that shows the averages to a box plot. The x-axis boundaries have been described in the legend (page 18).

“The superimposed box plot is drawn in red, and the x-axis boundaries of the groups occur at OD₆₀₀ values of 0-1, 1-2, 2-3 and 3-4.”

Figure 4C – E are not cited in the text

Our answer: Thank you for the comments. We have cited these figures in the text of the revised manuscript (page 8) as follows:

“By conducting numerical simulations using this model, we determined the critical density and heterotopic distribution in the presence of a high antibiotic concentration, which qualitatively agree with the experimental results (Figs. 4B-E and S6).”

Figure 5 – I don't get the statistics here, why the error bar only on the black bars and why bars at all?

Our answer: Thank you for the comments. The results presented using white bars in Fig. 5A and Fig. 5B were calculated by simulation. When the input parameters were unchanged, the calculation results were the same each time. We have added explained this further in the legend of Fig. 5 (page 21).

Figure 6 – Why the total community as inset, which is hard to read?

Our answer: Thank you for the comments. The R bacteria that did not easily die affected the total growth rate of the community, making it less obvious in the early stage of the culture. Moreover, since the green fluorescence of the labeled R bacteria was weak, the reading error of the number of the R bacteria was also greater than that of the N bacteria. Therefore, we thought that the growth rate of the N bacteria was more accurate than the total growth rate of the community. The total growth rate results for the community are shown in the illustrations to reflect the consistency of the changes in the growth rate of the the N bacteria. Because the illustration shows this trend, we believe that its inclusion is useful in this regard.

Figure 7: See fig 2B – why don't use box or violin plots here?

Our answer: Thank you for the comments. We have changed the dot plot showing the averages to a box plot. The x-axis boundaries have been described in the legend (page 23) as follows:

“The x-axis boundaries of the superimposed box occur at OD_{600} values of 0.5-1, 1-2, 2-4, 4-8, 8-16 and 16-32.”

References: Some informations – volume, pp (especially for the PNAS publications) are lacking

Our answer: Thank you for the comments. We have completed the information in the Reference section in the revised manuscript.

REVIEWERS' COMMENTS:

Reviewer #1 (Remarks to the Author):

In their revised manuscript „Spatial cooperation in a population of mutually beneficial bacteria enhances the colonial antibiotic resistance” the authors mostly address reviewer comments on presentation of the data, clarification of figures, placement of text sections, context and depth of some introductory and discussion paragraphs. In my opinion they addressed all comments satisfactorily with just a few minor issues remaining/arising through the revision. I thus suggest a minor revision of the following points.

Please add a method section on the added statistical tests, incl. justification of the used t-tests regarding normal distribution and variance homogeneity of the tested data in Fig. 5 and S8. And add the test for Fig. S9

Data availability: While the Nature Research Journals do not strictly require deposition of data in online available repositories, they strongly suggest it instead of only stating they are available upon request from the authors. I completely support this suggestion, but ultimately that's for the editor and journal to decide.

Fig. 6 and supp. Fig 11: Either add a note in the Figure descriptions what the numbers “1”, “2” and “3” under the three different trajectories mean or remove them if not necessary. I was wondering a while what they mean, but as far as I understood they are “just” three different exemplary samples.

Supp L94: change “mutually beneficial mutualism” to “mutually beneficial interaction” or similar.

Reviewer #2 (Remarks to the Author):

My comments have been addressed in the revised manuscript by Lingjuin Li et al. and I would recommend acceptance of the manuscript.

Reviewer #3 (Remarks to the Author):

My previous general comment: (..)For both points I found it hard to decipher what is new here and what are the key differences to the earlier work cited here as well (e.g. ref 25). Several introductory and discussion parts are found in the results section. (...)

Authors answer: “We have modified the Introduction and Discussion sections, removing some general discussions and adding more specific information.”

New general comments:

The authors did an overall very good job in revising the paper; the readability is much enhanced and the topic and the results are much clearer presented now. One general comment regarding the structure, especially the “discussion” part:

I recommend to reformat/rephrase (i.e. shorten and clarify to the main/overall messages you transport) and rename the current discussion part to the conclusion part. This also would imply to rename the results part to “results and discussion”, since there is a lot of discussion in there. A few specific comments on that will follow below.

"For the microfluidic chip used in ref 25, the gradient is stable when observing the bacterial behaviors, since there are source and sink wells in the device. A short observation period was used to determine the MIC value of the bacteria while the concentration gradient of the substance formed by the chip used in ref 25 was stable and not changed by bacteria. In our studies, a mutualistic community composed of two bacteria was used to observe the characteristics of the community. In addition, our device only had source wells. The concentration gradient and the activity of the bacteria would change with time, since a much higher bacterial density was used in this study compared with that used in Ref 25. Our system is more similar to natural conditions in that the bacterial mutualistic community aggregates and communicates with the outside world through limited contact surfaces (source) and the bacteria create a chemical gradient within the community by themselves. Based on the referee's suggestion, we have moved some of the information from the Results section to the Introduction and Discussion sections. Some of the methods described in the supplementary material have been added to the text. In addition, the dot plot showing the average value was changed to a box plot with statistical description. Finally, the order of the numbers in the supplementary material was arranged in the order of appearance in the main text."

New Comment: Many thanks for clarification. It would be nice if you would include the main message here also (briefly!) in the results section where it is appropriate (i.e. where you describe the outcome of the chip)

Original comment: Discussion – L 354 ff

The discussion reads more like a conclusion and contains more or less just general points. Some of the discussion is in the results part and in general the results are not too much discussed. Also, only few references are cited in a field which emerged from 2010 on and should provide more for discussion than just the eight cited references.

Authors answer: Thank you for the comments. We have modified the Discussion section (page 11) and

added additional references as follows:

"We observed that an initial OD600 1-2 was needed for growth in the 1600- μ m culture system, whereas a much higher initial bacterial density (such as an OD600 8-16 for the 800- μ m culture system)

was needed for a smaller culture system to resist the antibiotics present in the medium. These densities are much higher compared with those observed in MIC studies [25] using a similar microfluidic system and is smaller than that of formed biofilms in other microfluidic systems (approximately $3 \times 10^9 \sim 10^{10}$ cells/mL) [15,30]."

"The spatial structures observed in our experiments are different from those obtained in previous models or experiments [28,31]. This difference may be caused by the closure of the community growth space and the complexity of interactions between the populations."

New Comment:

I have difficulties with the changes applied to the discussion. It now is awkward to read and difficult to get the parts; initially I had some more fundamental changes in mind – but as Reviewer 1 remarked, the initial structure of the authors where a lot of results are initially discussed (i.e. in the results section) makes somehow sense. That would imply that the authors move the above parts which are currently in the discussion part (lines 339 to 344 and 352 to 355) to the results section and rename the results section to results and discussion. The discussion section can be renamed to conclusion part (if the journal style allows that) or just be the final paragraph in the results/discussion part.

Original reviewers comment to Figure 5 – I don't get the statistics here, why the error bar only on the

black bars and why bars at all?

Authors answer: Thank you for the comments. The results presented using white bars in Fig. 5A and Fig. 5B were calculated by simulation. When the input parameters were unchanged, the calculation results were the same each time. We have added explained this further in the legend of Fig. 5 (page 21).

New comment: Would it be possible to introduce dot plots instead of bar graphs? Then the statistical information would not be completely lost in the bar graphs. Especially then it would also get clearer, which part includes 3, 4 or 5 replicates. You could think of using one dot or one line (or stick to the white bars) then for the simulation results.

REVIEWERS' COMMENTS:

Reviewer #1 (Remarks to the Author):

In their revised manuscript, "Spatial cooperation in a population of mutually beneficial bacteria enhances the colonial antibiotic resistance" the authors mostly address reviewer comments on presentation of the data, clarification of figures, placement of text sections, context and depth of some introductory and discussion paragraphs. In my opinion they addressed all comments satisfactorily with just a few minor issues remaining/arising through the revision. I thus suggest a minor revision of the following points.

Please add a method section on the added statistical tests, incl. justification of the used t-tests regarding normal distribution and variance homogeneity of the tested data in Fig. 5 and S8. And add the test for Fig. S9

Our answer: Thank you for the comments. We added "Statistics and Reproducibility" with general information on how the statistical analyses of the data were conducted, and general information on the reproducibility of experiments in the revised manuscript.

Data availability: While the Nature Research Journals do not strictly require deposition of data in online available repositories, they strongly suggest it instead of only stating they are available upon request from the authors. I completely support this suggestion, but ultimately that's for the editor and journal to decide.

Our answer: Thank you for the comments. We chose figshare as the suitable data repository for our data and the link has been provided in the reply to editor.

Fig. 6 and supp. Fig 11: Either add a note in the Figure descriptions what the numbers "1", "2" and "3" under the three different trajectories mean or remove them if not necessary. I was wondering a while what they mean, but as far as I understood they are "just" three different exemplary samples.

Our answer: Thank you for the comments. We added a description of these numbers in the legend in the revised manuscript.

Supp L94: change "mutually beneficial mutualism" to "mutually beneficial interaction" or similar.

Our answer: Thank you for the comments. We changed it in the revised manuscript.

Reviewer #2 (Remarks to the Author):

My comments have been addressed in the revised manuscript by Lingjuin Li et al. and I would recommend acceptance of the manuscript.

Reviewer #3 (Remarks to the Author):

My previous general comment: (..)For both points I found it hard to decipher what is new here and what are the key differences to the earlier work cited here as well (e.g. ref 25). Several introductory and discussion parts are found in the results section. (...)

Authors answer: “We have modified the Introduction and Discussion sections, removing some general discussions and adding more specific information.“

New general comments:

The authors did an overall very good job in revising the paper; the readability is much enhanced and the topic and the results are much clearer presented now. One general comment regarding the structure, especially the “discussion” part:

I recommend to reformat/rephrase (i.e. shorten and clarify to the main/overall messages you transport) and rename the current discussion part to the conclusion part. This also would imply to rename the results part to “results and discussion”, since there is a lot of discussion in there. A few specific comments on that will follow below.

Our answer: Thank you for the comments. Due to the magazine's requirements for the format, we have to retain the original structure.

“For the microfluidic chip used in ref 25, the gradient is stable when observing the bacterial behaviors, since there are source and sink wells in the device. A short observation period was used to determine the MIC value of the bacteria while the concentration gradient of the substance formed by the chip used in ref 25 was stable and not changed by bacteria. In our studies, a mutualistic community composed of two bacteria was used to observe the characteristics of the community. In addition, our device only had source wells. The concentration gradient and the activity of the bacteria would change with time, since a much higher bacterial density was used in this study compared with that used in Ref 25. Our system is more similar to natural conditions in that the bacterial mutualistic community aggregates and communicates with the outside world through limited contact surfaces (source) and the bacteria create a chemical gradient within the community by themselves. Based on the referee’s suggestion, we have moved some of the information from the Results section to the Introduction and Discussion sections. Some of the methods described in the supplementary material have been added to the text. In addition, the dot plot showing the average value was changed to a box plot with statistical description. Finally, the order of the numbers in the supplementary material was arranged in the order of appearance in the main text.”

New Comment: Many thanks for clarification. It would be nice if you would include the main message here also (briefly!) in the results section where it is appropriate (i.e. where you describe the outcome of the chip)

Our answer: Thank you for the comments. We have added the description of this section in the revised manuscript (page 4).

“Compared with the chip used in other study, our device only had source wells and the concentration gradient would change with the bacteria growth. Our system is more similar to natural conditions in that the bacterial mutualistic community aggregates and communicates with the outside world through limited contact surfaces (source) and the bacteria create a chemical gradient within the community by themselves²⁵.”

Original comment: Discussion – L 354 ff

The discussion reads more like a conclusion and contains more or less just general points. Some of the discussion is in the results part and in general the results are not too much discussed. Also, only few references are cited in a field which emerged from 2010 on and should provide more for discussion than just the eight cited references.

Authors answer: Thank you for the comments. We have modified the Discussion section (page 11) and added additional references as follows:

“We observed that an initial OD600 1-2 was needed for growth in the 1600- μm culture system, whereas a much higher initial bacterial density (such as an OD600 8-16 for the 800- μm culture system) was needed for a smaller culture system to resist the antibiotics present in the medium. These densities are much higher compared with those observed in MIC studies [25] using a similar microfluidic system and is smaller than that of formed biofilms in other microfluidic systems (approximately $3 \times 10^9 \sim 10^{10}$ cells/mL) [15,30].”

“The spatial structures observed in our experiments are different from those obtained in previous models or experiments [28,31]. This difference may be caused by the closure of the community growth space and the complexity of interactions between the populations.”

New Comment:

I have difficulties with the changes applied to the discussion. It now is awkward to read and difficult to get the parts; initially I had some more fundamental changes in mind – but as Reviewer 1 remarked, the initial structure of the authors where a lot of results are initially discussed (i.e. in the results section) makes somehow sense. That would imply that the authors move the above parts which are currently in the discussion part (lines 339 to 344 and 352 to 355) to the results section and rename the results section to results and discussion. The discussion section can be renamed to conclusion part (if the journal style allows that) or just be the final paragraph in

the results/discussion part.

Our answer: Thank you for the comments. Due to the magazine's requirements for the format, we have to retain the original structure.

Original reviewers comment to Figure 5 – I don't get the statistics here, why the error bar only on the black bars and why bars at all?

Authors answer: Thank you for the comments. The results presented using white bars in Fig. 5A and Fig. 5B were calculated by simulation. When the input parameters were unchanged, the calculation results were the same each time. We have added explained this further in the legend of Fig. 5 (page 21).

New comment: Would it be possible to introduce dot plots instead of bar graphs? Then the statistical information would not be completely lost in the bar graphs. Especially then it would also get clearer, which part includes 3, 4 or 5 replicates. You could think of using one dot or one line (or stick to the white bars) then for the simulation results.

Our answer: Thank you for the comments. We have converted the bar graphs (Fig 5a and Fig 5b) to dot-plot format in the revised text.